# Non-volatile optical switch of resistance in photoferroelectric tunnel junctions

Xiao Long[1], Huan Tan[1], Florencio Sánchez [1], Ignasi Fina [1✉] & Josep Fontcuberta [1✉]

In the quest for energy efficient and fast memory elements, optically controlled ferroelectric memories are promising candidates. Here, we show that, by taking advantage of the imprint electric field existing in the nanometric $BaTiO_3$ films and their photovoltaic response at visible light, the polarization of suitably written domains can be reversed under illumination. We exploit this effect to trigger and measure the associate change of resistance in tunnel devices. We show that engineering the device structure by inserting an auxiliary dielectric layer, the electroresistance increases by a factor near $2 \times 10^3$%, and a robust electric and optic cycling of the device can be obtained mimicking the operation of a memory device under dual control of light and electric fields.

[1] Institut de Ciència de Materials de Barcelona (ICMAB-CSIC), Campus UAB, 08193 Bellaterra, Catalonia, Spain. ✉email: ifina@icmab.es; fontcuberta@icmab.cat

With rapid development of information technology and artificial intelligence, building energy efficient memory devices is one of the main objectives of the scientific community. Among the different strategies, memristor are attracting attention because they can act as artificial synapses able to simulate the excitation/depression process, fundamental to build bionic-like energy-efficient neural networks[1–4]. Ferroelectric tunnel junctions (FTJs) are prime memristor candidates[5–12]. In FTJs, voltage-induced polarization switching produces a change in the electrostatic energy landscape[13] resulting in important changes of resistance [called electroresistance (ER)], that can be induced by modest electric fields[14]. Dual optical and electrical manipulation of resistance state is of interest for neuromorphic vision sensors or optoelectric neuromorphic devices[15] and the optical control of resistance states in FTJ has been proposed as the ultimate energy efficient and ultrafast control mechanism[16].

Exclusive phenomena related to the presence of ferroelectric polarization and its interaction with light, such as large photo-striction[17] and the electric control of photovoltaic response[18], have attracted interest. For instance, photostriction in ferro-electric/magnetic heterostructures has been used to change magnetic properties by light[19,20] and memory cells based on electrically controlled and polarization-dependent photo-conductance have been designed[21–23]. Among them the optical control of polarization, namely, optical polarization switching, would be of the highest interest to achieve the fast control of resistance in memristive components based on ferroelectric materials[16]. The control of ferroelectric polarization by the pho-toabsorption and concomitant carriers generation in ferroelectrics was studied in the 1980s[24], and investigations of integration of ferroelectric materials in photo-storage applications were reported[25,26]. Nowadays, the interest on light-control of ferroic interaction is undergoing a renaissance[27]. Recently, it has been shown that the population of polar domains with different orientations and concomitant domain walls, in ferroelectric multidomain single crystals (BaTiO$_3$, BTO), can be modulated by suitable coherent illumination[28,29] or ultrafast light pulses[30]. Polarization switching via light absorption at semiconducting electrodes (for instance, bidimensional MoS$_2$)[31,32] or subsidiary devices[33,34] has been reported, and it has been used to control device resistance by light and electric field in a three terminal ferroelectric field-effect device[35,36]. Exploitation of polarization reversal by light absorption within the ferroelectric remains rather unexplored[37,38].

Here, we aim to achieve optical polarization switching and to gain optical control of HIGH/LOW resistance states (HRS, LRS) of FTJs, avoiding to rely on the photoabsorption in 2D systems (i.e. MoS$_2$)[36], whose integration with ferroelectric films is chal-lenging. We shall exploit a radically different approach, that relies on two concepts. First, the amplitude of the polarization (P) of a ferroelectric semiconducting single crystalline epitaxial film shall depend on the amount of available free charges[24] and conse-quently it can be modulated or eventually suppressed by intern-ally or externally (at the electrodes, at the ferroelectric/electrode interface or by surface adsorbates) generated photocharges[37–40]. Second, growth conditions[41–43] and electrostatic boundary conditions[44,45] during thin film deposition may lead to the existence of a so-called imprint electric field ($E_{IMP}$) that favors a particular direction of the emerging polarization P of the films. For instance, in Fig. 1a, as-grown $E_{IMP}$ downwards favors $P_{DOWN}$ ferroelectric domains, and a therefore a macroscopic $P_{DOWN}$ polarization. Still, the polarization can be switched from $P_{DOWN}$ to $P_{UP}$ by an applied electric field (E) larger than $E_C$ and $E_{IMP}$ ($E > E_C + |E_{IMP}|$, Fig. 1b). The suppression of P by internal pho-tocarriers[39], reduces the height of the energy barrier between $P_{DOWN}/P_{UP}$ states and eventually washes it out (Fig. 1c). After

suppressing photoconversion by switching off the illumination, the photogenerated carriers will ultimately recombine, and the polar state will be restored. As $E_{IMP}$ shall be still present, the $P_{DOWN}$ macroscopic polarization will be recovered (Fig. 1d). $E_{IMP}$ is thus instrumental for the ferroelectric to keep tract of the initial state and P reversal under illumination shall only occur if in the initial state P was antiparallel to $E_{IMP}$. If the ferroelectric layer is integrated in a tunnel device, polarization reversal leads to a large change of resistance between low and high resistance states (LRS and HRS; respectively) (Fig. 1e). By the same token, if transparent metallic electrodes are used, optical-induced polarization reversal from $P_{UP}$ to $P_{DOWN}$ will translate into resistance switch from LRS to HRS, as illustrated in Fig. 1f.

Here, we report on the polarization and resistance switch of a ferroelectric tunnel junction by visible light and we show how the device can be cycled to store information, which is subse-quently read by measuring the tunnel current across the device. In a first realization, an ultrathin epitaxial ferroelectric BTO layer is sandwiched between bottom and top metallic electrodes, La$_{2/3}$Sr$_{1/3}$MnO$_3$ (LSMO) and platinum (Pt), respectively. This heterostructure is used to elucidate the presence of $E_{IMP}$ and its direction, and to demonstrate the light-induced (blue light) polarization reversal. Next, the ER is measured. Data analysis allows to disentangle different contributions to ER, namely the contribution related to the ferroelectric polarization reversal, which is prevalent at low-voltage and high writing speed, superimposed to the ionic motion contribution, which is domi-nant in the opposite limits. Measurements under illumination clearly reveal that the resistive switching associated to polariza-tion reversal is very sensitive to photon absorption. Finally, we show that the electric and optical responses of the device are radically improved by engineering the interface between BTO and the LSMO bottom electrode. Indeed, by inserting a nanometric dielectric layer (SrTiO$_3$, STO) between the bottom LSMO elec-trode and the ferroelectric BTO barrier, a larger ER (increased by a factor near $2 \times 10^3$%) and a robust light-induced ON to OFF (LRS to HRS) resistance switch is observed allowing to demon-strate a dual electric-optical control of resistance in a FTJ memory element.

## Results

**Polarization switching by combined action of light and $E_{IMP}$ electric field.** We first demonstrate the switching of the polar-ization direction by visible light, in ultrathin 4 nm epitaxial BTO films grown using pulsed-laser deposition (PLD)[46]. The BTO layer is deposited on a LSMO (30 nm) conducting film grown on a (001) STO substrate in a single process (see the "Methods" section). X-ray diffraction $\theta-2\theta$ scans of the samples, confirm the excellent quality of the films and its [001] texture (Supplementary Note 1 and Supplementary Fig. 1). LSMO film is used as bottom electrode and it is grounded in all reported experiments, unless otherwise indicated.

The polarization direction of the as-grown film, inferred by piezoelectric force microscopy (PFM) (Fig. 2a) is found to be $P_{DOWN}$, indicating the presence of $E_{IMP}$ pointing downwards. It will be shown below that $E_{IMP} \approx -2.8$ MV cm$^{-1}$, which roughly corresponds to voltage imprint ($V_{IMP}$) $\approx -1.0$ V. By application of bias voltage ($V = \pm 8$ V) to the microscope tip (Supplementary Note 2 and Supplementary Fig. 2), $P_{UP}$ (black) and $P_{DOWN}$ (yellow) domains are defined (Fig. 2a), showing the electric control of ferroelectric polarization. The observation of near-zero piezoelectric response and 180° phase shift between domains of opposite polarity in PFM images (Supplementary Notes 3 and 4 and Supplementary Figs. 3, 4) strongly supports their ferroelectric nature[47,48]. PFM images have subsequently been collected after

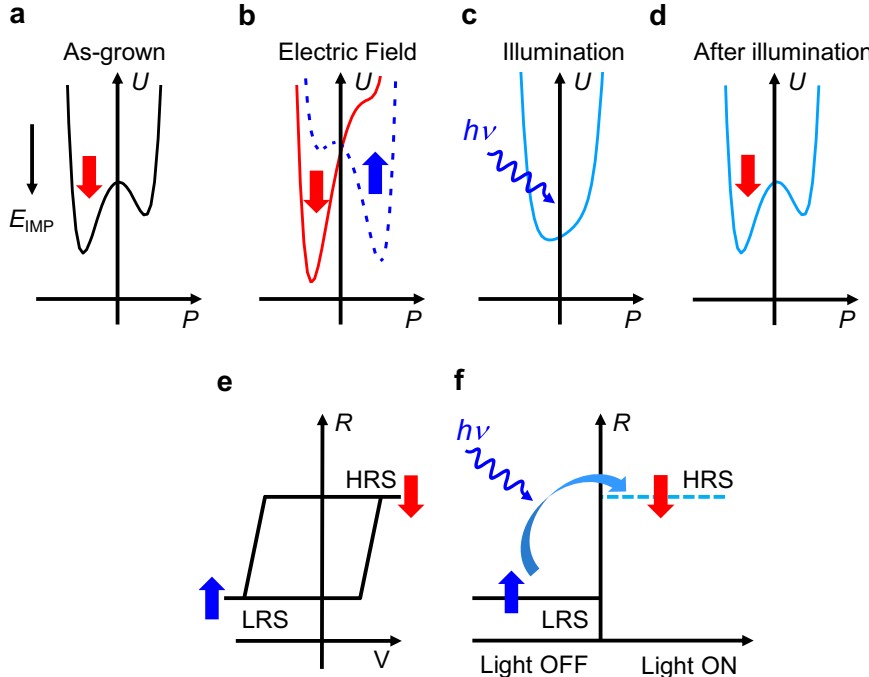

**Fig. 1 Polarization in a ferroelectric film with $E_{IMP}$, photoabsorption and its impact on polarization and tunnel ER.** Energy ($U$) of $P_{UP}$ (blue arrows) and $P_{DOWN}$ (red arrows) domains in a ferroelectric film, in presence of $E_{IMP}$ parallel to $P_{DOWN}$. **a** In the as-grown state. **b** Under external applied electric field larger or smaller than the coercive field ($E_C^+ - E_{IMP}$) and ($E_C^- + E_{IMP}$), respectively, polarization reversal occurs. **c** Under illumination, the polarization and the double potential energy well are suppressed. **d** After illumination, macroscopic polarization emerges along a direction dictated by pre-existing $E_{IMP}$. **e** Electroresistance (ER) loop measured in dark, in a ferroelectric tunnel junction as a function of a $V_W$. Loop is shifted towards $V < 0$, illustrating the presence of $E_{IMP}$. **f** Low-resistance state (LRS) to high-resistance state (HRS) switching promoted by optically induced polarization reversal.

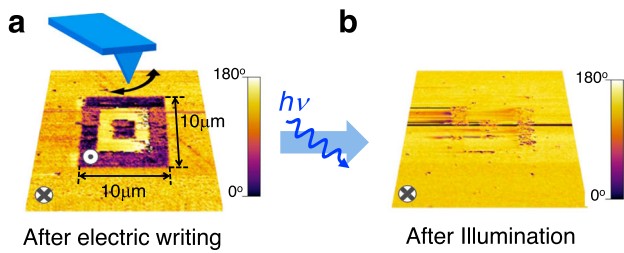

**Fig. 2 Polarization switching by combined action of light and $E_{IMP}$ electric field.** PFM phase-maps (30 µm × 30 µm) of BTO film. **a** $P_{DOWN}$ and $P_{UP}$ regions were written by applying voltage to the tip of −8 or +8 V within a 10 µm × 10 µm area in yellow and black regions, respectively. **b** After illumination (blue laser, 10 min) $P_{UP}$ domains are switched back.

illuminating the sample with a blue laser in dark conditions ($\lambda = 405$ nm). As shown by data in Fig. 2b, illumination has prompted $P_{UP}$ switching to $P_{DOWN}$. Control experiments indicated that, in dark, $P_{UP}$ states remain stable, without any obvious change in PFM contrast in at least 6 h (Supplementary Note 5 and Supplementary Fig. 5), implying that spontaneous polarization back-switching is not relevant. Therefore, data in Fig. 2 demonstrate that light-absorption assisted by $E_{IMP}$, promotes polarization switching of one of the polarization states ($P_{UP}$, in our BTO film). In the following, we shall exploit this phenomenon to demonstrate optical control of resistance in BTO FTJs.

**Electric field and light control of ER of BTO FTJs.** We turn now to the ER of BTO tunnel barriers on LSMO, using Pt as top electrode (LSMO/BTO/Pt) (sketch in Fig. 3a). First, we aim at identifying the different mechanisms contributing to the ER in

these FTJs. In Fig. 3a, we show an illustrative $I(V)$ curve recorded in dark (black line). It can be appreciated that the $I(V)$ curve displays two current peaks indicated by dashed lines for increasing positive applied voltage at: $V^+_{C\text{-}LOW} \approx 3$ V [best seen in Fig. 3a (inset)] and $V^+_{C\text{-}HIGH} \approx 13$ V. In Fig. 3b, we show the junction resistance ($R$) recorded by varying the writing voltage ($V_W$). Data were collected at remanence, i.e. after application of a suitable writing triangular voltage pulse of amplitude $V_W$ and duration $\tau_{write} = 1$ ms (see the "Methods" section). Black symbols in Fig. 3b correspond to $R(V_W)$ data collected in dark. Loops are stable upon further cycling (Supplementary Note 6 and Supplementary Fig. 6). Two distinguishable voltages, at $V^+_{C\text{-}LOW} \approx 3$ V and $V^+_{C\text{-}HIGH} \approx 13$ V, can be identified. At $V^+_{C\text{-}LOW}$ and $V^+_{C\text{-}HIGH}$ there is a sudden increase of resistance; these voltages closely coincide with those where current switching peaks are observed in the $I(V)$ curves (Fig. 3a). We focus first our attention on the changes of resistance occurring at the $V_W > 0$ region. The existence of two critical voltages ($V^+_{C\text{-}LOW}$ and $V^+_{C\text{-}HIGH}$) indicates the presence of different switching mechanisms. One can expect that these coexisting mechanisms might have different time response. The fastest mechanism could be related to a pure electronic process, i.e. ferroelectric switching. Figure 3c shows the $R(V_W)$ loop recorded using $\tau_{write} = 100$ µs. A coercive field, $E_C \approx 8.0$ MV cm$^{-1}$ (corresponding to $V_C \approx 3.2$ V) and an $E_{IMP} \approx -2.8$ MV cm$^{-1}$ (corresponding to $V_{IMP} \approx -1.0$ V) can be extracted. Note that although $E_C$ is comparable to the breakdown field in polycrystalline samples[49], time-dependent current measurements under constant electric stress and endurance data do not agree for dielectric breakdown at $V^+_{C\text{-}LOW} < V_W < V^+_{C\text{-}HIGH}$. Instead, equivalent characterization collected for $V_W > V_{C\text{-}HIGH}$ is consistent with the proposed ionic motion scenario although the presence of soft dielectric breakdown after successive cycling cannot be disregarded (Supplementary Notes 7 and 8 and

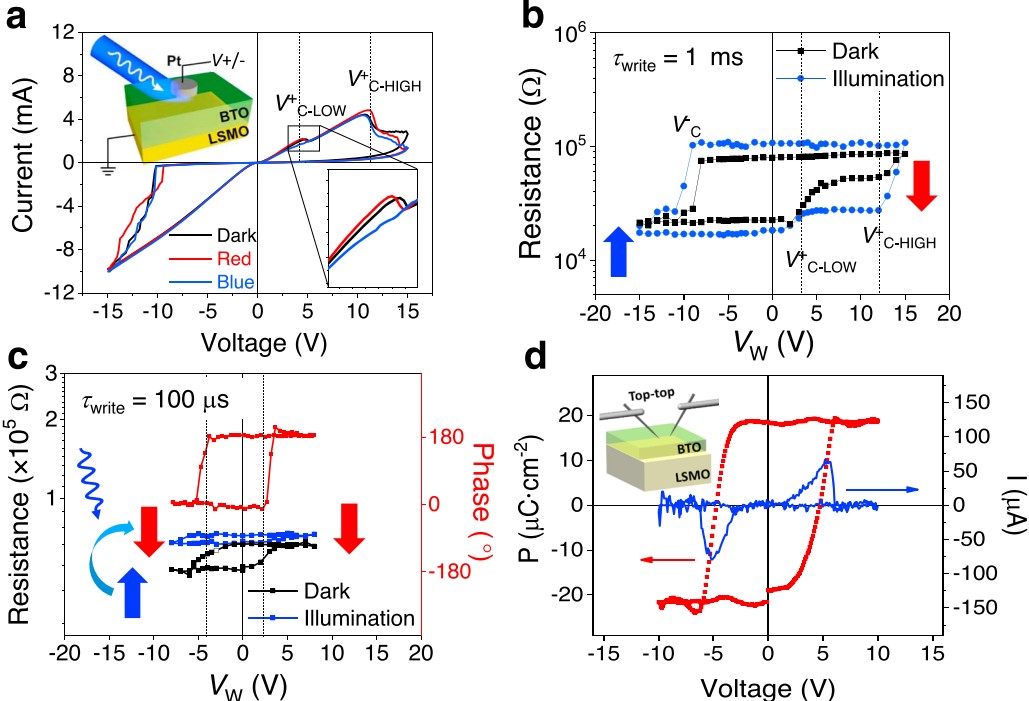

**Fig. 3 Electroresistance under illumination. a** $I(V)$ loops following the −15 to +15 to −15 V path collected in dark and under illumination (red and blue laser). Inset top-left: Schematic of the measuring configuration. Inset bottom-right: Zoom of the $I(V)$ loops around $V^+_{C-LOW}$. **b** $R(V_W)$ loops collected using $\tau_{write} = 1$ ms in dark and under (blue) illumination. Arrows indicates polarization direction. **c** $R(V_W)$ loops collected using $\tau_{write} = 100$ μs in dark (black symbols) and under (blue) illumination (blue symbols). The PFM phase-voltage hysteresis loop (red symbols) of the sample (right-bottom axis). Vertical dashed lines indicate the $E_C$ in PFM loop, coinciding with $V^+_{C-LOW}$ and $V^-_{C-LOW}$ in the $R(V_W)$ loop. **d** $P(V)$ and $I(V)$ loop and sketch of the measuring configuration.

Supplementary Figs. 7 and 8). The PFM phase loop (recorded using $\tau_{write} = 100$ μs) (Fig. 3c) shows coercive and imprint voltages which are closely coinciding with those extracted from $R(V_W)$ data (vertical dashed lines). The $I(V)$ curves in the low-voltage region ($V_W < V^+_{C-HIGH}$) can be well fitted by an electron tunnel transport model across polarization modulated tunnel barriers (Supplementary Notes 9 and 10, Supplementary Figs. 9, 10 and Supplementary Table 1), providing additional support to the ferroelectric origin of the ER in this voltage region. $P(V)$ and $I(V)$ loops (Fig. 3d) collected in top–top configuration (see the "Methods" section) further indicate the ferroelectric nature of the sample. Switching current peaks appear at 5.6 V, which is about twice $V_C$ observed in the $R(V_W)$ loop, as expected from an antiserial connection of ferroelectric capacitors[50]. Polarization value is also in reasonable agreement with bulk values and similar to values obtained in BTO films grown in the same nominal conditions and similar thickness[51]. Therefore, we conclude that the $R(V_W < V^+_{C-HIGH})$ loops collected at short $\tau_{write}$ (<1 ms) are governed by ferroelectric polarization switching, which give rise to a LRS ($P_{UP}$) to HRS ($P_{DOWN}$) resistive switching and vice versa upon $V_W$ cycling.

The presence of two critical voltages ($V^+_{C-LOW}$, $V^+_{C-HIGH}$) in $R(V_W)$ data collected at $\tau_{write} = 1$ ms in Fig. 3b, allows stabilizing three different resistance states: low (LRS ≈ 30 kΩ) state, intermediate (HRS ≈ 50 kΩ) state at $V^+_{C-LOW}$ and high state, labeled $HRS^{ION}$, ($HRS^{ION}$ ≈ 90 kΩ), at $V^+_{C-HIGH}$. Interestingly, when using a faster writing time, $\tau_{write} < 500$ μs, the LRS and HRS resistance states are visible in the $R(V_W)$ loops but the $HRS^{ION}$ state is absent in all tested junctions (Supplementary Notes 11 and 12 and Supplementary Figs. 11, 12). One might suspect that a slower resistive switching mechanism, not responsive to fast writing pulses (≤500 μs), is related to an ionic migration process[52,53]. Taking $\tau_{write} = 500$ μs as the diffusion time of

ions across the BTO film thickness to reach the interface under $V^+_{C-HIGH} \approx 13$ V, a mobility ≈ $2.5 \times 10^{-9}$ cm$^2$ V$^{-1}$ s$^{-1}$ is estimated. This estimate is within the range of values reported for the motion of oxygen vacancies motion in BTO at room temperature[54] and it is orders of magnitude smaller than typical electron mobility in similar oxides (≈1 cm$^2$ V$^{-1}$ s$^{-1}$), suggesting defect migration is the most probable mechanism[55–59].

The change of resistance occurring at $V_W < 0$ (after the $V_W > 0$ excursion) and particularly the abrupt reduction of resistance observed at $V^-_C \approx -10$ V (Fig. 3a, b), makes indiscernible the electronic and ionic transport channels. Analogous abrupt change of resistance at $V_W < 0$ was reported in other BTO-based FTJs[53,60] and it was interpreted in terms of ferroelectric domain avalanches stimulated by the asymmetric voltage drop across the junctions.

Now, we turn to the electrical characterization of the FTJs under illumination conducted by using blue ($\lambda = 405$ nm, $E = 3.06$ eV) and red ($\lambda = 638$ nm, $E = 1.94$ eV) lasers, near and well below BTO bandgap ($E_g = 3.3$ eV)[61], respectively. A FTJ capacitor was completely exposed to the laser beam, as sketched in Fig. 3a. The $I(V)$ characteristics recorded under red and blue light are shown in Fig. 3a. Strikingly, it is observed that the current peak at $V^+_{C-LOW}$ is suppressed when the junction is illuminated by the blue laser. In contrast, $I(V)$ curves recorded under red laser and dark are almost identical. These results indicate that the supression of the polarization switching, which we recall is at the origin of the observed current peak at $V^+_{C-LOW}$, results from photocarriers generated at the BTO layer by the blue laser. This is confirmed by the absence of polarization switching by red light, observed by PFM (Supplementary Note 13 and Supplementary Fig. 13). Thermal effects, in a 4 nm BTO film, would have similar effects for both wavelengths, in contrast to observation, thus suggesting to have a minor role on the $I(V)$ photoresponse. The contribution of LSMO photoconductance has

been verified, being it negligible and thus suggesting that LSMO contribution in the ferroelectric switching polarization is not relevant (Supplementary Note 14 and Supplementary Fig. 14). In Fig. 3b, the $R(V_W)$ loop collected ($\tau_{write} = 1$ ms) under blue light is shown. It can be observed that, under illumination, the ER at $V^+_{C-LOW}$ is largely suppressed. Instead, $V^+_{C-HIGH}$ remains and signals the onset of a large change of resistance. These observations indicate that the LRS to HRS switching resulting from polarization reversal is suppressed by illumination. Instead, the ionic-like conduction channel at $V^+_{C-HIGH}$ persists and governs the ER, giving rise to the HRS$^{ION}$ state. Figure 3c shows that the resistance of the HRS obtained by voltage control ($V_W <V^+_{C-HIGH}$) is coincident with that measured under illumination (blue symbols), which confirms that polarization is switched from $P_{UP}$ to $P_{DOWN}$ by the blue laser as schematized in Fig. 3c. An identical behavior is found when measuring the ER in dark and under illumination of junctions with BTO barriers of different thicknesses ($\approx 2$ and 5 nm) (Supplementary Note 10 and Supplementary Fig. 10). As the polarization has been already switched by light, the $R(V_W)$ loops collected with $V_W < V^+_{C-HIGH}$ under blue light, display a very reduced ER, which is in agreement with the absence of the switching peak in the $I(V)$ data under blue light (Fig. 3a). All loops collected under light in the $V_W < V^+_{C-HIGH}$ voltage range and using the same $\tau_{write}$, show a similar ER suppression (Supplementary Note 15 and Supplementary Fig. 15). Moreover, it is observed that some junctions display non-reversible photoresponse (Supplementary Note 15 and Supplementary Fig. 15), as reported in ferroelectric single crystals and thin films[62,63], and attributed to charge trapping. It thus follows that, to take full advantage of the photoresponsive ER in FTJs, the ionic transport channel should be mitigated or suppressed. It will be shown in the following that engineered interfaces allow to overcome this limitation.

**Enhanced optical control of ER in hybrid dielectric/ferroelectric junctions.** Aiming at suppressing ionic contributions (possibly oxygen vacancies) to the ER and eventually obtaining fully polarization-controlled tunnel barrier suitable for optical resistive switching, we have introduced a dielectric blocking layer (STO, 1 nm thick) between the LSMO bottom electrode and the BTO layer (LSMO/STO/BTO), as sketched in Fig. 4a. It is expected that this dielectric layer could have a double benefit. First, limiting ionic motion across the LSMO/BTO interface and second, blocking the flow of photocarriers through the circuit while allowing photocarriers in BTO to screen $P$, thus the polarization-related tunnel barrier or Schottky barrier at BTO interfaces will remain sensitive to photocarriers. The STO thickness (1 nm) is selected to be thin enough to allow tunneling across the whole device. Figure 4b displays the $R(V_W)$ loop collected in a junction in this hybrid structure LSMO/STO/BTO ($\tau_{write} = 100 \mu s$) in dark (black symbols) and under illumination (blue symbols). Loops are stable upon further cycling (Supplementary Note 16 and Supplementary Figure 16). Four important features can be observed. First, the ER in dark increases by a factor near $2 \times 10^3\%$ compared to data in Fig. 3c, and it is robust under constant cycling (Supplementary Note 17 and Supplementary Fig. 13). Indeed, the change of resistance is of about 500% in the LSMO/STO/BTO (Fig. 4b) while it was of $\approx 30\%$ in LSMO/BTO junctions (Fig. 3c). This enhancement results from the larger asymmetry of the electronic potential profile across the device. A similar approach had been used earlier to improve ER in ferroelectric tunnel barriers[9,64]. Moreover, whereas two critical ($V^+_{C-LOW}$ and $V^+_{C-HIGH}$) fields, related to polarization reversal and ionic motion, respectively, were observed in LSMO/BTO junctions, only the $V^+_{C-LOW}$ remains apparent in the $R(V_W)$ loops of

the LSMO/STO/BTO hybrid junctions. This observation confirms that the STO layer contributes to suppress the ionic channel and the HRS$^{ION}$ state[64] (Supplementary Note 18 and Supplementary Fig. 18). Second, it is found that the $E_{IMP}$ increases by about a factor 2 (from $-1$ V in the LSMO/BTO junction (Fig. 3c) to $-2$ V in the LSMO/STO/BTO (Fig. 4b)). The $E_{IMP}$ increase also contributes to stabilize a HRS ($P_{DOWN}$), that consistently with the notation in previous section we still denote as HRS, without compromising the ER magnitude. Third, under illumination, the polarization-related ER is completely suppressed and the system remains in the HRS. This observation is crucial as it indicates that, although ferroelectric polarization is suppressed, $E_{IMP}$ induces non-ferroelectric (paraelectric) polarization parallel to $P_{DOWN}$ under illumination. And fourth, the $R(V_W)$ loop after illumination (open symbols) is very similar to the one measured before illumination (black symbols), which indicates the robustness of the ferroelectric controlled ER. To confirm the effect of light on the LSMO/STO/BTO structure, PFM images of LSMO/STO/BTO were collected in dark and after illumination. First, after writing with $V_W = \pm 8$ V ferroelectric domains with opposite polarity, a PFM phase image was collected (Fig. 4c, left panel). The PFM contrast among $P_{UP}/P_{DOWN}$ regions is obvious although smaller than in the PFM of the LSMO/BTO sample (Fig. 2b) due to the more relevant charging effects[48] in the LSMO/STO/BTO structure (see Supplementary Note 3 and Supplementary Fig. 3). After illumination using the blue laser, PFM images were collected again. It can be appreciated in Fig. 4c (right panel) that the contrast between $P_{UP}$ and $P_{DOWN}$ has significantly decreased compared with the image collected in dark, indicating light-induced $P_{UP}$ polarization suppression. Contrast profiles shown in Fig. 4c (bottom panel) provide a direct evidence of $P_{UP}$ suppression by (blue laser) light. $P(V)$ and $I(V)$ loops (Fig. 4d) collected in top–top configuration (see the "Methods" section) further indicate the ferroelectric nature of the sample. Switching current peaks appear at 4.4 V, which is close to two times $V_C$ ($\approx 8.4$ V) from $R(V_W)$ loop, as expected[50]. The reduced polarization compared with the LSMO/BTO sample is expectedly smaller due to different electrostatic boundary conditions of the BTO layer.

The LSMO/STO/BTO hybrid structure allows a reversible (light-on/light-off) ER response. Therefore, combined voltage–illumination protocols can be used to handle the non-volatile resistance state (memory) of the device. To illustrate this prospect, we show in Fig. 5 the sequence of stimuli ($V_W(\tau_{write} = 100 \mu s)$, blue-laser light) acting on the FTJ and the subsequent resistance states. In Fig. 5a, a HRS ($P_{DOWN}$) is first written by applying a $V_W = +8$ V pulse (step 1); after a LRS ($P_{UP}$) state is written by a $V_W = -8$ V pulse (step 2); the HRS ($P_{DOWN}$) is recovered (step 3) after illumination of the sample while the sample is unbiased. Afterwards, it is observed that applying $V_W = +8$ V the HRS ($P_{DOWN}$) state does not change as expected (step 4). The LRS ($P_{UP}$) is recovered (step 5) by applying $V_W = -8$ V. Thus LRS optically switch to HRS. Reciprocal results are shown in Fig. 5b. In this case, it is observed that HRS ($P_{DOWN}$) does not switch under illumination. Data in Fig. 5 illustrates in the clearest way how light-induced polarization reversal can be used to control the memory state of the device. Similar results are obtained in different devices (Supplementary Note 19 and Supplementary Fig. 19).

Figure 6a demonstrates the remarkable retention of the electrically and/or optically written LRS and HRS states of the device. The resistance of the ON/OFF states, electrical written using voltages pulses of $-8$ V for LRS and $+8$ V for HRS, are recorded as a function of the time elapsed after electrical writing. Figure 6a also illustrates the retention of the HRS induced by light, after writing with $-8$ V. All light-induced resistive switching experiments reported above, were performed using fixed

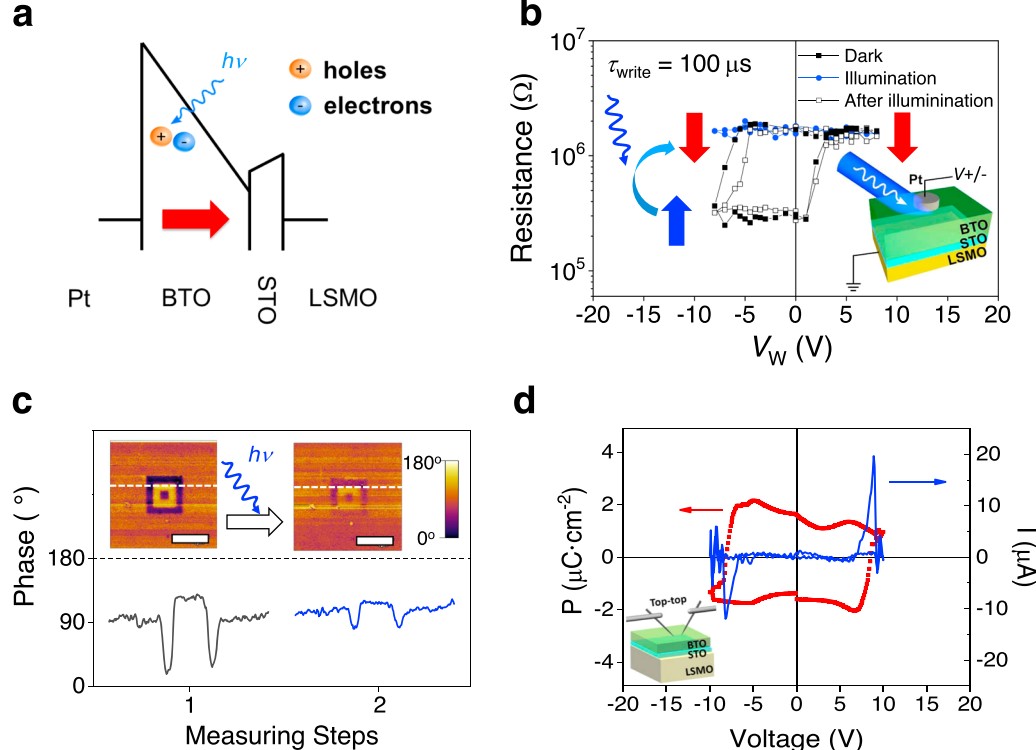

**Fig. 4 Electroresistance of dielectric/ferroelectric heterostructures under illumination. a** Sketch of the layer stacking with a polarized BTO barrier.
**b** $R(V_W)$ loop collected using $\tau_{write} = 100\ \mu s$ in dark (black solid and open circles) and under illumination (blue symbols). **c** Out-of-plane PFM phase images obtained after writing $P_{DOWN}$ and $P_{UP}$ domains (applied voltage to the tip of +8 V (inner yellow region) and −8 V (black region)), respectively (left image) and after illuminating the LSMO/STO/BTO sample (right image). Scale bars correspond to 10 μm. Bottom, phase-profiles along the white dashed lines in the corresponding images. **d** $P(V)$ and $I(V)$ loop and sketch of the measuring configuration.

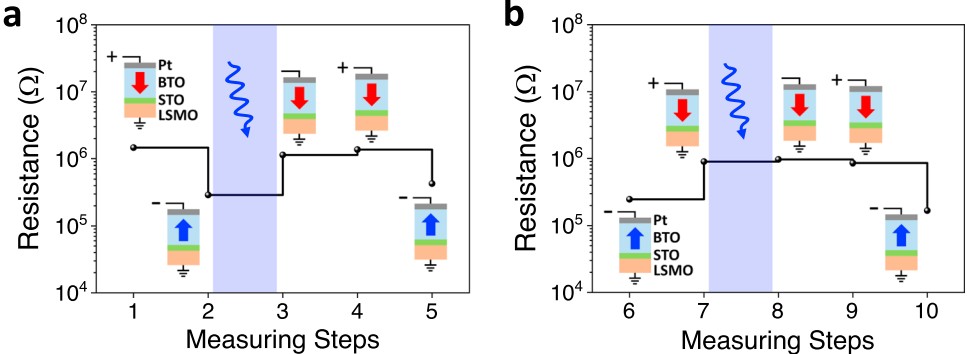

**Fig. 5 Electro-optical control of resistance.** Resistance measured under different electric and optical stimulus sequence. Initial states at LRS and HRS in **a** and **b**, respectively. Notice (blue shaded areas) that $P_{UP}$ states are reversed by light and resistance switches from LRS to HRS, whereas $P_{DOWN}$ states are insensitive to light. In this example: $V_W = \pm 8$ V. Optical illumination is marked with symbol. Arrows indicate polarization direction.

illumination conditions (see "Methods" section) that were selected to completely switch the LRS to HRS state. Fine-tuning (memristive-like) of optical switch is achieved by reduction of laser power and illumination time. Figure 6b illustrates the LRS-to-HRS resistance switching dependence on illumination time when the laser power is reduced to 2 W/cm². Interestingly, under these conditions, the resistance of the device can be continuously varied from LRS to HRS. Naturally, the switching is faster when increasing the laser power (Supplementary Note 20 and Supplementary Fig. 20).

## Discussion
We have shown that the combined action of $E_{IMP}$ and optical stimulus is used to switch ferroelectric polarization and

concomitantly, the resistance state in a FTJ. Robust optical-electric switching is found in the LSMO/STO/BTO (dielectric/ferroelectric) heterostructures, demonstrating that the optical control of resistance is non-volatile. These hybrid dielectric/ferroelectric heterostructures appear to mitigate ionic-like conducting channels across the FTJs and improving ER and stability compared to the junctions based on LSMO/BTO bilayer. The results presented here demonstrate a novel dual stimulus (voltage/light) control of the memory state of ferroelectric junctions. The optical switch sets resistance state as stable as the electrically set; in addition, appropriate selection of illumination time and power allows fine-tuning of the final resistance state, i.e. memristive behavior. The observed optical control, based on polarization switching, relies on the existence of an $E_{IMP}$ that is

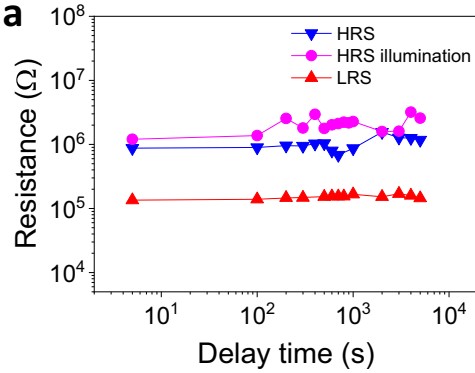
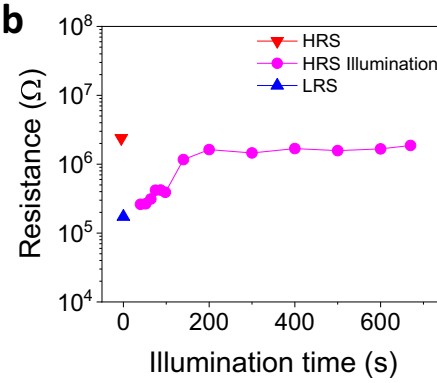

**Fig. 6 Retention and optical modulation of resistance. a** Retention of the voltage ($\tau_w = 100\,\mu s$) and light written HRS states and LRS state in a junction on STO//LSMO/BTO/STO/Pt. **b** Evolution of the resistance of the junctions under illumination, from LRS to HRS. Data (blue circles) are collected as function of illumination time using a laser power of 2 W/cm². Each plot also includes the resistance of the LRS and HRS measured in dark (up blue and down red, respectively). Arrows indicate the direction of the polarization of the ferroelectric barrier.

dictated by the BTO deposition conditions. Alternative sources can be envisaged to mimic $E_{IMP}$ and unbalance ad hoc $P_{UP}$ and $P_{DOWN}$ states without requiring changing the polarity by $V_W$. They may offer alternatives and may be more practical ways to achieve full optical control of polarization in FTJs. Pivotal for the operation of the device are the photocarriers required to suppress the ferroelectric polarization, here obtained by photoabsorption in BTO. The large gap of BTO implies that photobsorption is mainly dictated by defects in the film, of limited absorbance. It is clear that larger efficiency could be obtained by exploiting narrower gap ferroelectrics, such as BiFeO₃ or hexagonal manganites, both having bandgaps in the visible range.

## Methods

**Device preparation.** STO//LSMO (30 nm)/BTO (4 nm) and STO//LSMO (30 nm)/ STO (1 nm)/BTO (4 nm) samples were grown in situ by PLD. The heterostrutures were deposited in a single process. The LSMO layer was grown at 725 °C, at oxygen pressure of 0.1 mbar and 2 Hz of laser frequency. The BTO was grown at 700 °C, at oxygen pressure of 0.02 mbar and 5 Hz. Further experimental details can be found elsewhere[46]. Top Pt electrodes (20 nm thick) were deposited by sputtering through a stencil mask, allowing to obtain arrays of contacts of 20 and 7 μm diameter.

**Electrical characterization.** Electrical characterization was conducted by using the contact configuration shown in Fig. 3a. The bottom electrode (LSMO) was grounded while the $V(t)$ signal was applied to the top electrode (Pt). Current–voltage [$I(V)$] curves were measured by applying triangular $V(t)$ pulses with a maximum amplitude of 15 V. ER loops were collected by extracting the resistance at 0.5 V from $I(V)$ curves collected in the 0.5 V range measured after applying trapezoidal voltage pulses of amplitude $V_W$ and $\tau_{write}$, were $\tau_{write}$ is the rise time, plateau time and decay time of the trapezoidal signal (voltage pulse train described in Supplementary Note 21 and Supplementary Fig. 20). All measurements were done at room temperature using an AixACCT TFAnalyser2000 platform. Data have been obtained using contacts of 20 and 7 μm diameter. Data of STO/BTO sample in the manuscript refer to junctions with 20 μm diameter top electrodes. Fully consistent results are obtained using the smaller contacts (7 μm) (Supplementary Notes 22–24 and Supplementary Figs. 22–24).

**Ferroelectric characterization.** Ferroelectric $P(V)$ and $I(V)$ loops were performed by antiserial connection of two ferroelectric capacitors in the so-called top–top configuration (see sketches in Fig. 3d, d). In this configuration, $I(V)$ asymmetry is canceled-out. Needles directly in contact with the surface to minimize the contact area and thus the presence of leakage current added to polarization switching were used. The effective contact area was ≈80 μm². The leakage was subtracted following the procedure described in Supplementary Note 25 and Supplementary Fig. 25. All measurements were done at 5 kHz at room temperature using an AixACCT TFAnalyser2000 platform. Polarization was obtained from the current versus time ingratiation normalized to the area.

**PFM characterization.** PFM measurements were performed with an MFP-3D ASYLUM RESEARCH microscope (Oxford Instrument Co.), using the AppNano

silicon (*n*-type) cantilevers with Pt coating (ANSCM-PT-50). Scanned areas were 30 × 30 μm² and the electrically written regions were 10 × 10 μm².

**Illumination set-up.** Optical illumination during or between the electrical measurement was performed with blue laser ($\lambda = 405$ nm and $\lambda = 638$ nm lasers, with an illuminating power density of ~48.5 W cm⁻²) driven by a CPX400SA DC power (AimTTi Co.). At $\lambda = 405$ nm the optical transparency of the used Pt electrodes is ~15%[65,66]. Incidence angle ≈ 45° is fixed for all the measurements. In PFM and electrical measurements performed after illumination and shown in Figs. 2, 5, respectively, the sample was illuminated with power density of ~48.5 W cm⁻² and exposure time of 10 min. PFM experiments were conducted either before or after illumination.

**Reporting summary.** Further information on research design is available in the Nature Research Reporting Summary linked to this article.

## Data availability

The data of the figures of this paper and materials are available from the corresponding authors upon reasonable request.

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

## Acknowledgements

Financial support from the Spanish Ministry of Science, Innovation and Universities, through the "Severo Ochoa" Program for Centres of Excellence in R&D (FUNFUTURE, CEX2019-000917-S) and the MAT2017-85232-R (AEI/FEDER, EU), MAT2015-73839-JIN (MINECO/FEDER, EU), PID2019-107727RB-I00 (MINECO/FEDER, EU) projects, and from Generalitat de Catalunya (2017 SGR 1377) is acknowledged. I.F. acknowledges RyC Contract RYC-2017-22531 and Beca Leonardo from Fundación BBVA. X.L and H.T are financially supported by China Scholarship Council (CSC), respectively with nos. 201806100207 and 201906050014. The work of X.L. and H.T. have been done as a part of their Ph.D. program in Materials Science at Universitat Autònoma de Barcelona. We are extremely thankful to R. Solanas for his skillful operation of the PLD system. R. Silvestre-Anglada is acknowledged for assistance in $I(V)$ characteristics fittings.

## Author contributions

I.F and J.F. planned the study. H.T. in collaboration with X.L. performed PFM characterization. X.L. performed the electric characterization. F.S., I.F., and J.F. were responsible for sample preparation. X.L., I.F., and J.F. wrote the manuscript with inputs from all authors.

## Competing interests

The authors declare no competing interests.
