## [Peer Review File · Nature Communications]

REVIEWER COMMENTS

Reviewer #1 (Remarks to the Author):

The article title "Non-volatile optical switch of resistance in photoferroelectric tunnel junctions" by Long et al. reported photo-induced switching of ferroelectric polarization for optical control the electrically written data.

The results in the current work is interesting for the community and could have nice applications, if the observed effect can be reproduced reliably and different technical issues are properly addressed in the manuscript. I have few suggestions/comments for clearing the confusions that a reader might have and some technical data that would be useful for the community.

1. In the article, it has been claimed that the optical reversal of ferroelectric domains is an unexplored territory. However, optical reversal of ferroelectric domains has been reported in several works the past. For instance, the MoS₂/BTO/SRO structures, "Optical control of polarization in ferroelectric heterostructures" (Nature Communications volume 9, Article number: 3344 (2018)) or Photo-induced ferroelectric switching in perovskite CH₃NH₃PbI₃ films"(Nanoscale, 2017, 9, 3806) or for ferroelectric liquid crystals where ferroelectric domain switching by illumination has been shown to be due to optically induced charge carrier modulation either at the semiconductor/ferroelectric interface or due to ionic motions in combination with a photovoltaic field. Also, modification of ferroelectric polarization by optically induced charge carriers are shown in several studies as well, for example, for PZT/LSMO heterostructures ("Morphology-dependent photo-induced polarization recovery in ferroelectric thin films" (Appl. Phys. Lett. 111, 092902 (2017)) where the morphology of the PZT film determines the relaxation of the FE domains. Also photoexcited carrier generation and photoinduced resistive switching has been shown in the LSMO electrode itself where observation of photoconductivity (at high voltage) even in the presence of interfacial LaTiO₃ layer is shown that suggests the role of oxygen vacancy diffusion in the observed photoinduced effect in presence of an electric field. Light and gate voltage driven diffusion of oxygen vacancy in these heterostructure has been shown to alter the orbital occupancy of eg electrons in the LSMO layer (Electric field control of photoinduced effect in La_{0.7}Sr_{0.3}MnO₃/LaTiO₃/SrTiO₃ heterostructure. Phys. Rev. B 100, 115119 (2019). I suggest the authors include some important previous works in the discussion in this respect, clearly stating the actual novelty of the findings and how the previously reported mechanisms agree/oppose their findings.

From the technical side, I suggest addition of following results in the manuscript:

1. The non-volatility data of the optically written domains should be presented with a retention measurement together with a comparison of the electrically written data stability.
2. Also, reproducibility of the switching voltages over several cycles of operation and over different samples would be interesting to have a proper understanding of the effect.
3. A dependence of the observed photoswitching effect on the thickness of the ferroelectric and the dielectric layer could provide interesting information for the readership.
4. A write-pulse time dependent Ron/off plot together with light-pulse intensity and duration dependence of domain switching could provide interesting information on the contribution of electronic and ionic origin of the effect.
5. A junction size dependence of the effect.

Reviewer #2 (Remarks to the Author):

The authors report on the Electroresistance (ER) effect in tunnel junctions of ferroelectric (FE) BaTiO₃. The paper shows the ER has two contributions, from the FE polarization (which creates an asymmetry

on the tunnel barrier changing the resistance upon polarization switching) and from ionic (defect) motion across the barrier. The two mechanisms give rise to two different switching fields and two different time constants. The important effect of ionic or defect migration in ferroelectric tunnel junctions has been reported by several authors recently [Beach's group, see <https://www.nature.com/articles/nmat4134>] and not so recently [M. Bowen, <https://aip.scitation.org/doi/10.1063/1.2345592>]. This effect has also been investigated by the present group on similar devices made of ferroelectric HfZrO₄.

In the present manuscript, the ion migration effect has been mitigated by adding a 1 nm layer of SrTiO₃ to the junction, as in Wang et al (ref. 9 of the present manuscript). The novelty of this work is the increased control of the ER by illumination, since absorption of light by defect states can supply additional carriers to screen the polarization. The very serious issue of the paper is that the authors are applying 8V for films below 10 nm-thick, that is a field of about 1 MV/cm, which is exactly the breakdown field of BTO thin films (see <https://d-nb.info/1048647420/34>) (see figure attached)

So there is the possibility that what the authors are seeing is breakdown or some pre-breakdown effect. In order to discard that the authors should

1) Report more details about the device stability. The authors refer to previous work to report the details on the device. They mention ref. 43, which seems to be a typo and I assume they mean ref. 45. In that reference the authors mention that the top electrodes are made of Pt, that the diameter of the electrodes is 250 μm^2 and that there are 300 electrodes on the samples. The size of the electrode seems very large and one wonders if other mechanisms such as thermal filament formation are not present in the material. In the suppl. Information there are measurement of 5 devices/electrodes. How many cycles do the devices survive? How many electrodes do not work or directly break? This could give an idea of how close to breakdown the working conditions are.

2) In addition, one also expects a more convincing proof of ferroelectricity in this ultra-thin barriers and clear evidence that one of the two switching mechanism is, indeed, ferroelectric switching. Contrast retention of 20 mins could be caused by electrochemical reactions at the surface under the huge electric field applied to the sample. If a filament is formed, the ER effect is due to a formation/destruction of a few atoms junction, which can also be very fast. The difficulty of PFM to distinguish real ferroelectricity from other surface charging effects is well known (Kalinin et al.)

Moreover, and as a less critical point of concern, in the supplementary information, the authors include a fit of the tunneling barrier assuming that the transport is by direct tunneling through the barrier (a Brinkman model, with tunneling current through a trapezoidal potential barrier. Have they tried other models?

The authors should also look at the recently work by Xi Wang et al. (2019 Mater. Res. Express 6 046307) in which they show ER effects in BaTiO₃ tunnel junctions with ON/OFF current ratios of ~ 170 for 3.2 nm barrier and $\sim 12\,500$ for the 4.8 nm barrier. This enhancement was reported to be due to (i) the improved ferroelectric-modulation of the barrier profile and (ii) the evolution of the transport mechanism with increasing BTO thickness (suppression of tunneling in the 4.8 nm-thick BTO layer and thermally-activated thermionic injection becoming active in the thicker layers). Could this not be playing a role also in this case: STO could simply play the same role as a thicker tunnel barrier with a different transport mechanism. Notice that a couple of unit cells of STO can even become polar next to a ferroelectric.

In short, there are too serious open questions for me to be able to advise the publication of this work in a high impact journal like Nat. Comm.

Reviewer #3 (Remarks to the Author):

This is certainly an interesting manuscript in which the combination of above band-gap illumination and imprint, inherent to the ferroelectric layer within the grown heterostructure, facilitate a "back-switching" event, such that domains poled into an "up state" can be back switched into a "down-state" using light alone (the electric field also needed is already inherently present and is responsible for the imprint). A mechanism as to how this is facilitated through photo-excited internal screening is discussed, as are complications associated with ionic transport and the fact that such issues may be obviated by incorporating a blocking layer - in this case an ultrathin STO film.

I have no major objection to publication in Nat. Commun., but would recommend a few things (mixture of very minor and quite minor):

- (i) the label on figure 3c indicates that the pulse write time is 100 milliseconds, rather than 100 microseconds - this is really important and had me confused for a while. I think it is just a font issue;
- (ii) I would be more explicit about the red / blue lasers being below / above the expected band-gap for the BTO...at the moment it is implied, but I don't think it is stated clearly in the main text at least;
- (iii) while I get a sense for the vision as to how the imprint+light illuminated generation of free carriers might work, I am aware that there are other aspects to the electrostatics that are clearly there, but seem not to be so important to the light-induced switching phenomenon: perhaps a more open discussion including, for example, the role of the photo-induced carriers over and above screening available due to the conducting electrodes / free surfaces in air and to the charge-carrying point defects (oxygen vacancies) that we know are present from the ion-current characterisation. In other words, can the authors explore a little more what the photo-induced carriers are able to do, in terms of screening, that other screening mechanisms that should be fully expected seem unable to facilitate.

REVIEWER COMMENTS

Reviewer #1 (Remarks to the Author):

The article title “Non-volatile optical switch of resistance in photoferroelectric tunnel junctions” by Long et al. reported photo-induced switching of ferroelectric polarization for optical control the electrically written data.

The results in the current work is interesting for the community and could have nice applications, if the observed effect can be reproduced reliably and different technical issues are properly addressed in the manuscript. I have few suggestions/comments for clearing the confusions that a reader might have and some technical data that would be useful for the community.

R#1-C0. In the article, it has been claimed that the optical reversal of ferroelectric domains is an unexplored territory. However, optical reversal of ferroelectric domains has been reported in several works the past. For instance, the MoS₂/BTO/SRO structures, “Optical control of polarization in ferroelectric heterostructures” (Nature Communications volume 9, Article number: 3344 (2018)) or Photo-induced ferroelectric switching in perovskite CH₃NH₃PbI₃ films” (Nanoscale, 2017, 9, 3806) or for ferroelectric liquid crystals where ferroelectric domain switching by illumination has been shown to be due to optically induced charge carrier modulation either at the semiconductor/ferroelectric interface or due to ionic motions in combination with a photovoltaic field. Also, modification of ferroelectric polarization by optically induced charge carriers are shown in several studies as well, for example, for PZT/LSMO heterostructures (“Morphology-dependent photo-induced polarization recovery in ferroelectric thin films” (Appl. Phys. Lett. 111, 092902 (2017)) where the morphology of the PZT film determines the relaxation of the FE domains. Also photoexcited carrier generation and photoinduced resistive switching has been shown in the LSMO electrode itself where observation of photoconductivity (at high voltage) even in the presence of interfacial LaTiO₃ layer is shown that suggests the role of oxygen vacancy diffusion in the observed photoinduced effect in presence of an electric field. Light and gate voltage driven diffusion of oxygen vacancy in these heterostructure has been shown to alter the orbital occupancy of eg electrons in the LSMO layer (Electric field control of photoinduced effect in La_{0.7}Sr_{0.3}MnO₃/LaTiO₃/SrTiO₃ heterostructure. Phys. Rev. B 100, 115119 (2019).

I suggest the authors include some important previous works in the discussion in this respect, clearly stating the actual novelty of the findings and how the previously reported mechanisms agree/oppose their findings.

R#1-C0 Response. *We thank the Referee for drawing our attention to some recent literature of photoinduced effects in ferroelectrics and more precisely on polarization switching. We absolute agree that careful update of information and referencing is essential, and in this sense, we acknowledge very much the Referee for reminding us some references. Indeed, the mentioned APL and the Nanoscale manuscript are connected to some of the findings reported in our manuscript, as we detail below. They have been included in the revised version as Refs. 37 and 38. The suggested reference dealing with MoS₂ (Nature) was already included [old Ref 30, now 31].*

The APL and Nanoscale manuscripts report on the switching of a FE layer (PZT and halide) under illumination, and the observed effects (polarization switching) are claimed to be related to photoabsorption within the FE. In contrast, in the Nature paper, it is claimed that the observed polarization switching occurs due to photo-absorption into the 2D narrow gap semiconductor acting as electrode, rather than in the FE. The PRB paper, the opinion of authors is that the relevant

absorption process occurs at the metallic electrode (LSMO) and this triggers the polarization switching (new Ref. 23).

Obviously the physics and the material's requirement are totally different in these two scenarios. Our approach is more connected to the first case.

However, in our opinion, the crucial point is that in none of those examples, and to the best of our knowledge in any previous report, photo-induce polarization switching is exploited as a tool to modulate the responsivity (a dramatic change of electroresistance) in a functional device.

In that sense we claimed in the introduction that "Exploitation of polarization reversal by light absorption within the ferroelectric remains rather unexplored" but we didn't write "the optical reversal of ferroelectric domains is an unexplored territory" as the Referee mentions. In our opinion, we were very careful wording the sentence.

Action: The following references have been added:

- Appl. Phys. Lett. 111, 092902 (2017)
- Nanoscale 9, 3806 (2017)
- Phys. Rev. B 100, 115119 (2019)

To emphasize novelty of approach and results some few sentences have been rewritten/added in pag. 2 and 3.

- In page 2, we add: "on electrically-controlled and polarization-dependent photoconductance have been designed.²¹⁻²³"
- In pag 3 we rephrase "Exploitation of polarization reversal by light absorption within the ferroelectric remains rather unexplored.^{37, 38}"
- In pag. 3 we add: and "First, the amplitude of the polarization P of a ferroelectric semiconducting single crystalline epitaxial film shall depend on the amount of available free charges²⁴ and consequently it can be modulated or eventually suppressed by internally or externally (at the electrodes, at the ferroelectric/electrode interface or by surface adsorbates) generated photocharges.³⁷⁻⁴⁰"

R#1-C1. The non-volatility data of the optically written domains should be presented with a retention measurement together with a comparison of the electrically written data stability.

R#1-R1. We have collected retention data by using PFM and electrical resistance as shown below. Data show excellent polarization retention, and together with data in Figure 3, indicate a remarkable electrically written resistance stability.

- 1) Figure R1 shows PFM measurements collected after writing with ($\pm 8V$), either immediately (Original) (a) and after waiting for 1 h and 3 h (b,c). It can be observed that the phase contrast is preserved (d) and the dark regions slightly expand with time (d) as expected from the presence of E_{imp} .

Figure R1. PFM phase images collected after electrical lithography with +8 (bright) and -8V (dark) for the LSMO/BTO sample. (a) Data collected freshly after writing (original); (b) after 1 h and (c) after 6 h. (d) Line scans (red lines in panels a-c) of the phase-contrast of images in (a,b,c). (e) Dependence on delay time since electrical writing of relative area ratio of dark/bright areas [=area_{dark/bright}/(total area)]. Lines are guides for the eye.

2) *Figure R2 shows different resistance states of ON/OFF states versus delay time after electrical writing using +8 V for HRS and -8 V for LRS voltage pulses. Figure R2 also includes the HRS induced by light, after writing with -8 V, and its dependence on the time elapsed between the optical writing and the reading. Resistance state is stable in all cases.*

Figure R2. Retention of the voltage ($\tau_w = 100 \mu s$) and light written HRS states and LRS state in a junction on STO//LSMO/BTO/STO/Pt. Resistance state is stable in all cases.

Action: Retention results of Figure R2 are now included in the text (pg.11) of the revised manuscript (new Figure 6) and PFM images of Figure R1 data are included in the SI Fig. S4.

R#1-C2. Also, reproducibility of the switching voltages over several cycles of operation and over different samples would be interesting to have a proper understanding of the effect.

R#1-R2. Data on some other few samples had already been obtained prior to submission, although only a couple of them were included in the manuscript. We attach here some data (Figs R3 and R4) for other junctions and samples, illustrating the observed effects are reproducible.

Figure R3. Illustrative examples of the reproducibility of the major ER loops in two different junctions in a STO//LSMO/BTO/Pt sample as a function of the writing time: (a) 250 ms and (b) 100 μ s. Data show the reproducibility of ER loops over several cycles of operation.

Figure R4. Illustrative examples of the reproducibility of the major ER loops in two different junctions in a STO//LSMO/BTO/STO/Pt sample for different writing time: (a) 1 μ s, (b) 10 μ s, (c) 2 ms and (d) 250 ms. Data show the reproducibility of ER loops over several cycles of operation.

Action: Results are described in the text and experimental data (Figs R3 and R4) are included in the SI Fig. S5 and S11.

R#1-C3. A dependence of the observed photoswitching effect on the thickness of the ferroelectric and the dielectric layer could provide interesting information for the readership.

R#1-R3. We had already included the dependence of the LRS and HRS on thickness in Supplementary Information Fig. S5. Now, following the advice of the Referee, we have updated this Figure S5 by including the values of the light-induced HRS induced state, for junctions of several BTO thickness. We also include here for completeness. The dependence of the light-induced HRS induced state on STO is an interesting open question out of the scope of the present work that we must address in the near future.

Figure R5. Resistance dependence on BTO thickness of the voltage (downward red triangle) and light (blue circles) written HRS states and LRS (upward blue triangle) state in STO//LSMO/BTO/STO/Pt samples.

Action: Updated Fig. S7 in SI including the thickness dependence of the illumination effect.

R#1-C4. A write-pulse time dependent Ron/off plot together with light-pulse intensity and duration dependence of domain switching could provide interesting information on the contribution of electronic and ionic origin of the effect.

R#1-R4. This is an interesting new research activity that, stimulated by the Referee comment, we have now addressed. Results are described in the text and experimental data are included in the SI.

A. We collected ER loops (major) to explore the dependence of the ON/OFF states on the duration of the writing pulses. Some data was already displayed in SI Fig. S8 and S9. It was observed that the two resistance states LRS and HRS are visible irrespective of τ_{write} , while the HRS^{ION} only appears at $\tau_{write} > 500 \mu s$.

B. We have collected ER loops (minor) under different illumination power for LSMO/STO/BTO/Pt devices. Some illustrative results are displayed in Figure R6. It can be observed that for 2 and 12 W/cm^2 the ER is completely suppressed, as for the 48.5 W/cm^2 laser power density reported in the manuscript. Smaller laser power is not possible in our experimental set up.

Figure R6. ER loops collected at $V_{max} = 8 V$ for different laser power. ER is completely suppressed for power as low as (a) 2 W/cm^2 and (b) 12 W/cm^2 , as for the 48.5 W/cm^2 laser power density reported in the manuscript.

C. A complementary set of new experiments has been done to determine how the resistance of the HRS state induced by light depends on illumination conditions (power and duration of light

pulses). Results, shown in Figure R7 below, indicate that illumination produces a gradual switching from LRS to HRS, which timescale is dictated by the power of the laser and the duration of the illumination step. We also include in this plot the resistance of the HRS and LRS states written in dark. It is remarkable the close coincidence of the final states (HRS) obtained by V-pulses and light-pulses.

The experimental observation that the light allows a fine tuning of the final state, may open new perspective that go much beyond the contents of this manuscript.

Figure R7. Evolution of the resistance of the junctions under illumination, from LRS to HRS. (a,b,c) depict data (blue circles) collected as function of illumination time when different laser power are used (2, 6, and 23 W/cm^2 , respectively). We also include in each plot, the resistance of the LRS and HRS measured in dark (up blue and down red, respectively).

D. Although referee did not emphasize on reproducibility of results in different devices, this is an important point that was not stressed in the first version of the manuscript. Some results are included in Figure R8 below, where we depict the resistance obtained after electrical and optical writing for several junctions. Although, as mentioned in the main manuscript, absolute resistance values display some junction-to-junction variability, the optical switch of resistance is systematic observed.

Figure R8. (a-d) Resistance states obtained sequentially after electrical and optical stimuli (as indicated) for several junctions, respectively. In this example: $V_w = \pm 8 \text{ V}$. Illumination is performed. Illumination is done using $\lambda = 405 \text{ nm}$ and power density of $48.5 \text{ W}/\text{cm}^2$.

Action: Results are described in the text and experimental data of Figures R7 and R8 are included in the SI Fig. S14 and S15. Panel R7a has been included in the main text (pg.11) as new Figure 6b.

R#1-C5. A junction size dependence of the effect.

R#1-R5. Although the masks limit the span of available contact sizes, data have been collected on junctions having circular electrodes of different size (7 and 20 μm of diameter). Data included in the early version of the manuscript correspond to 20 μm of diameter. Comparative data for 7 μm and 20 μm junctions are shown in Figure R9 and Figure R10 below. In Figure R9 we show data for STO//LSMO/BTO/Pt. The switching response is clearly observed using both electrodes (Figure R9(a)) and in the corresponding ER loops (Figure R9(b)). In Figure R10 we depict the ER loops collected in junctions of different size in STO//LSMO/STO/BTO/Pt heterostructure. In data in Figure R9 and R10 it can be appreciated that junction resistance and the ER are larger in the smaller junctions, as commonly found. This is typically related to the existence of non-switchable low-resistance channels. The fact that the effect is area dependent also disregard filamentary conduction as triggering mechanism of the found effect. In Figure R11, we show the light-induced electroresistance measured using $7 \times 7 \mu\text{m}^2$ electrodes, in dark and under illumination. The ER suppression under illumination is comparable to that obtained using the larger ($20 \times 20 \mu\text{m}^2$) electrodes.

Figure R9 Illustrative examples of the I-V curves (a) and ER (b) recorded in dark in junctions in STO//LSMO/BTO/Pt electrodes with diameter of 7 and 20 μm , as indicated.

Figure R10 Illustrative ER loops recorded in dark in different junctions (a,b) in STO//LSMO/STO/BTO/Pt having electrodes with diameter of 7 and 20 μm , as indicated.

Figure R11 Illustrative ER loops recorded in dark and under illumination electrodes with diameter of 7 and 20 μm of STO//LSMO/BTO/STO/Pt sample.

Action: Results are described in the text and experimental data are included in the SI Fig. 17, S18 and S19.

Reviewer #2 (Remarks to the Author):

R#2-CO. The authors report on the Electroresistance (ER) effect in tunnel junctions of ferroelectric (FE) BaTiO₃. The paper shows the ER has two contributions, from the FE polarization (which creates an asymmetry on the tunnel barrier changing the resistance upon polarization switching) and from ionic (defect) motion across the barrier. The two mechanisms give rise to two different switching fields and two different time constants. The important effect of ionic or defect migration in ferroelectric tunnel junctions has been reported by several authors recently [Beach's group, see <https://www.nature.com/articles/nmat4134>] and not so recently [M. Bowen, <https://aip.scitation.org/doi/10.1063/1.2345592>]. This effect has also been investigated by the present group on similar devices made of ferroelectric HfZrO₄.

R#2-RO. *We agree that the role of ionic motion in tunnel barriers and across metal/oxide or oxide/oxide interfaces, is not new. In fact, the manuscript already includes some illustrative relevant references (Refs 17-22) although obviously we have not attempted to cover the vast existing literature on the topic. In our opinion, inclusion of a longer list of references on examples of ionic contribution to modify barriers and interfaces, without a proper discussion on similarities or differences, would not provide much new information to readers. If such information were included, the focus of the reader on the novelties of the manuscript will be lost. We illustrate this objection using the examples provided by the Referee. The Bowen results (<https://aip.scitation.org/doi/10.1063/1.2345592>) involve using Cr as electrode on top of STO. Authors claim that Cr is very reactive and pulls oxygen from the barrier. We fully agree. We could also add that Co (also used in that manuscript) is also very reactive. In any case, from the material's science perspective is not a surprise that some transition metals can largely modify the barrier stoichiometry. In fact, this was also observed in the Beach's paper (<https://www.nature.com/articles/nmat4134>).*

R#2-CO. In the present manuscript, the ion migration effect has been mitigated by adding a 1nm layer of SrTiO₃ to the junction, as in Wang et al (ref. 9 of the present manuscript). The novelty of this work is the increased control of the ER by illumination, since absorption of light by defect states can supply additional carriers to screen the polarization. The very serious issue of the paper is that the authors are applying 8V for films below 10nm-thick, that is a field of about 1MV/cm, which is exactly the breakdown field of BTO thin films (see <https://d-nb.info/1048647420/34>) (see figure attached ????)

R#2-RO *We strength that ER is observed occurring in two steps, at two different voltage regimes (what we call V_{C-LOW} and V_{C-HIGH}). It is clear that V_{C-LOW} , occurring at $\approx +3V$ is closely coincident with the corresponding critical voltages observed in the piezo-loops. In contrast the V_{C-HIGH} occurs at larger voltages (10 -13 V) (depending on films). As the referee noticed, the corresponding electric field is comparable to the breakdown field of polycrystalline (Ba_{1-x}Sr_x)TiO₃ films, as extensively studied by Scott et al [Integrated Ferroelectrics. 1994. Vol. 4, pp. 61-84]. It is known that grain boundaries play a role on dielectric breakdown field E_{DB} , and thus in their absence it could be larger; however, we cannot exclude that the resistive switching observed at $\approx 10V$ may be related to some sort soft breakdown.*

Therefore, we should accept the comment of the Referee. The high field ER switching could be due to a soft breakdown. As there are many different possible origins for breakdown: intrinsic (electronic and thermal) or extrinsic (defect related); where some of them are related to ionic motion, we prefer to avoid this complex discussion which is beyond the scope of the manuscript.

Action: We acknowledge the comment by the Referee and we have included a comment on this issue in (Pag 6): "A coercive field, $E_c \approx 8.8$ MV/cm and an $E_{imp} \approx -2.8$ MV/cm (corresponding to $V_{IMP} \approx -1.0$ V) can be extracted. Note that E_c is below the breakdown field in polycrystalline samples.⁴⁷"

R#2-C1 So there is the possibility that what the authors are seeing is breakdown or some pre-breakdown effect. In order to discard that the authors should

1) Report more details about the device stability. The authors refer to previous work to report the details on the device. They mention ref. 43, which seems to be a typo and I assume they mean ref. 45. In that reference the authors mention that the top electrodes are made of Pt, that the diameter of the electrodes is $250 \mu\text{m}^2$ and that there are 300 electrodes on the samples. The size of the electrode seems very large and one wonders if other mechanisms such as thermal filament formation are not present in the material. In the suppl. Information there are measurement of 5 devices/electrodes. How many cycles do the devices survive? How many electrodes do not work or directly break? This could give an idea of how close to breakdown the working conditions are.

R#2-R1. First, we would like to comment on thermal filament formation. Results shown in Figures R9-R11 shown that the found ER scales with the area and that the optical suppression under illumination is present independently on the area of the measured device. The first set of results suggest that the observe effect is bulk rather than filamentary.

Regarding the question "How many cycles do the devices survive?" In Figures R3 and R4, we have shown the ER loop of several devices. It is observed that complete ER(V) loops can be recorded several times.

Regarding the question of "How many electrodes do not work or directly break?" The number of not working electrodes is $< 15\%$ on a given wafer. However, the absolute resistance values obtained among devices vary somehow as it can be inferred from Figures R4 and R8 for the LSMO/STO/BTO sample. Average among 10 measured electrodes is 114 ± 45 and $7 \pm 4 \times 10^4 \Omega$ for HRS and LRS, respectively (error bar corresponds to the standard deviation among obtained values). Although, resistance variation among different electrodes is significant, the HRS and LRS states are well distinguishable in all devices.

Regarding the retention. We have addressed this point by measuring the resistance of the LRS and HRS states in dark and under illumination for a period of time of a junction in STO//LSMO/BTO/STO/Pt. Resulting data, displayed in Figure R2 show that data are stable at least up to 7200s.

Data in Figure R8 demonstrates the repeatability of ON/OFF states induced by the combined actions of voltage and light in different devices.

Action: Figures R4 to R8 have been included in SI Fig. S11 and S14.

R#2-C2. In addition, one also expects a more convincing proof of ferroelectricity in this ultra-thin barriers and clear evidence that one of the two switching mechanism is, indeed, ferroelectric switching.

R#2-R2. We agree with the general statement made by the Referee about the contrast observed in PFM. We are well aware of the difficulties and possible pitfalls of the technique (as reviewed for instance in Vasudevan et al. *Applied Physics Reviews* 4, 021302 (2017); doi: 10.1063/1.4979015). The protocol used for PFM (writing / reading) was optimized as described below and included now in the Experimental section to minimized possible charging effects as in Ref. *Nanoscale* 9, 3806 (2017).

Probably more convincing is the observation that the piezoelectric loops – which are less prone to charging effects than the writing/reading, give coercive voltages coinciding with the critical voltages where the low-voltage resistive switching occurs (Fig. 3c).

On top of this observation, we are convinced that the optical suppression of the supposedly ferroelectric contribution to the electroresistance (Fig. 3c and 4b) and the corresponding suppression of the PFM contrast at illuminated regions leave little doubts. In perovskite oxides, we do not expect light-induced ion migrations, which is known to occur in other “softer” materials, such as halides [Nat. Commun. 24; 11683 (2016); S. Meloni et al. Nat. Commun. 7, 10334 (2016)]

Finally, the fact that PFM characterization shows that under illumination P-up domains reverse but not P-down is in full agreement with electroresistance measurements. We stress that ER data were collected using electrodes and thus surface adsorbates, whose presence is unavoidable in PFM, do not play a major role. The observed coincidence of switching voltages could hardly be a coincidence.

We end by mentioning that reversing polarization by light absorption at FE thin films has already been reported (Wang APL 111, 092902 (2017), although thicker PZT films were used. Here we exploit this effect and we go beyond by reporting the associated changes of resistance in tunnel devices

R#2-C3 Contrast retention of 20 mins could be caused by electrochemical reactions at the surface under the huge electric field applied to the sample. If a filament is formed, the ER effect is due to a formation/destruction of a few atoms junction, which can also be very fast. The difficulty of PFM to distinguish real ferroelectricity from other surface charging effects is well known (Kalinin et al.)

R#2-R3 We have performed additional experiments extending retention measurements in PFM until 6 h (see Figure R1), as sometimes done in literature (see for instance APL 111, 092902 (2017)). In Figure R12 below, we show PFM images collected after writing with different amplitudes (± 10 V, ± 8 V and ± 7 V) a region of the sample to define a standard $20 \times 20 \mu\text{m}^2$ square motif (sketched in Figure S2 in Supple. Information Fig. S2), and recording the PFM image (amplitude and phase) immediately after writing (fresh) and after 1h. This set of data is fully consistent, and again suggests that polarization contrast and polarization reversal is a genuine FE response rather than an effect of adsorbates.

Figure R12 The surface of the STO//LSMO/BTO sample ($20 \times 20 \mu\text{m}^2$) is polarized (written) by scanning the PFM tip biased by ± 10 V, ± 8 V and ± 7 V, as indicated. The PFM images shown are

the phase-map collected after 10 min and 1h. We notice that these experiments are done in open conditions, using the very same tip for writing and recording the subsequent PFM image.

Action: PFM characterization of Figure R1 data are included in the SI Fig. S4.

R#2-C4 Moreover, and as a less critical point of concern, in the supplementary information, the authors include a fit of the tunneling barrier assuming that the transport is by direct tunneling through the barrier (a Brinkman model, with tunneling current through a trapezoidal potential barrier. Have they tried other models?

R#2-R4. *The purpose of the fits was simply to verify if a tunnel model lead to: a) reasonable values of the energy barrier and b) if the expected exponential increase of the junction resistance with thickness was observed or not. As shown in Supp. Information S5, both objectives are clearly accomplished. A tunnel transport model accounts for the shape of the I-V curves. We have only systematically tested direct tunneling model. We have only made preliminary attempts to fit I(V) data using Fowler-Nordheim tunneling (FNT) and Thermionic injection (TI) following the equations described in Pantel et al. (Phys. Rev. B 82, 134105, 2010). It turns out that good fits can be obtained by using the TI model and the extracted barrier energies are almost identical (≈ 0.5 V) for both barrier sides. A barrier of ≈ 0.5 V for the BTO/Pt interface is much smaller than that expected from simple estimates of the barrier height on the basis of work function of Pt and electron affinity of BTO. We had therefore concluded that the fit was not meaningful and concentrated on the direct tunnel one. In an independent set of experiments on related samples, we have observed an exponential dependence of the junction resistance on the barrier width, which naturally fits into the expectations of direct tunneling (see SI Figures S4 and S5). The change of the barrier parameters is as expected for a ferroelectric tunnel barrier. This provide and additional support to the FE polarization reversal as the origin of the observed ER (at $|V| < V_{C-LOW}$)*

Action: A sentence clarifying this point has been included in Supplementary Information S6:

“Attempts to fit I-V characteristics using Fowler-Nordheim tunneling (FNT) and Thermionic injection (TI) following the equations described in Pantel et al.⁷ It turns out that good fits can be obtained by using the TI model and the extracted barrier energies are almost identical (≈ 0.5 V) for both barrier sides. A barrier of ≈ 0.5 V for the BTO/Pt interface is much smaller than that expected from simple estimates of the barrier height on the basis of work function of Pt and electron affinity of BTO. We had therefore concluded that the fits using FNT and TI models were not meaningful.”

R#2-C5. The authors should also look at the recently work by Xi Wang et al. (2019 Mater. Res. Express 6 046307) in which they show ER effects in BaTiO₃ tunnel junctions with ON/OFF current ratios of ~ 170 for 3.2 nm barrier and $\sim 12\,500$ for the 4.8 nm barrier. This enhancement was reported to be due to (i) the improved ferroelectric-modulation of the barrier profile and (ii) the evolution of the transport mechanism with increasing BTO thickness (suppression of tunneling in the 4.8 nm-thick BTO layer and thermally-activated thermionic injection becoming active in the thicker layers). Could this not be playing a role also in this case: STO could simply play the same role as a thicker tunnel barrier with a different transport mechanism. Notice that a couple of unit cells of STO can even become polar next to a ferroelectric.

R#2-R5. *The referee rises an interesting point. Which is the actual role of the inserted STO layer on the ER? We already mentioned in the manuscript that: “This enhancement results from the larger asymmetry of the electronic potential profile across the device. A similar approach had been used earlier to improve ER in ferroelectric tunnel barriers.”^{9, 55}. This statement was based on previous dedicated studies reported in literature. The selected Ref. 5 and 55, are examples of the statement above. To our understanding, the results of Wang et al. mentioned by the referee are another nice example of “modifications of the electronic profile across the device”. We have included the mentioned reference.*

We would like to stress, however, that in our case, the major role of the STO barrier is not to change the energy profile but to block the “ionic-like” channel in the electroresistance. Notice in Fig 4a, that V_{C-ION} is suppressed by insertion of the STO barrier. A similar approach has been recently shown to be also successful in limiting a related ionic contribution to ER in the celebrated HfO2 [Sulzbach et al. Adv. Funct. Mater. 2002638 (2020)]

R#2-C6. *In short, there are too serious open questions for me to be able to advise the publication of this work in a high impact journal like Nat. Comm.*

R#2-R6. *In our opinion, the referee main concerns are related to the nature of the observed resistive switching and the relative importance of polarization and ionic channels. We agree that these are important aspects. We provide robust evidences of the ferroelectric and non-ferroelectric nature on both channels and we demonstrate, that a capping layer mitigates to some extent, the latter.*

We are convinced that the novelty of our results is that light allows to control resistive switching, from LRS to HRS, in a way has not been reported before, and used this approach to trigger the functionality of a ferroelectric device. In this reply letter we hope we have provided extensive and conclusive details on reproducibility, size scaling, time dependence response writing pulses, time and power dependence on illumination protocol, that leave little doubts on the robustness of the results, understanding and fine experimental control. The additional information included in this Reply letter (Fig R7), demonstrating fine tuning of electroresistance by light may open new opportunities.

Action: *Panel R7a has been included in the main text (pg.11) as new Figure 6b.*

Reviewer #3 (Remarks to the Author):

R#3-C0 This is certainly an interesting manuscript in which the combination of above band-gap illumination and imprint, inherent to the ferroelectric layer within the grown heterostructure, facilitate a "back-switching" event, such that domains poled into an "up state" can be back switched into a "down-state" using light alone (the electric field also needed is already inherently present and is responsible for the imprint). A mechanism as to how this is facilitated through photo-excited internal screening is discussed, as are complications associated with ionic transport and the fact that such issues may be obviated by incorporating a blocking layer - in this case an ultrathin STO film.

R#3-R0. *We appreciate very much the positive view of the Referee on our manuscript and the elegant way in which the Referee summarizes our main findings.*

R#3-C1. I have no major objection to publication in Nat. Commun., but would recommend a few things (mixture of very minor and quite minor):

(i) the label on figure 3c indicates that the pulse write time is 100 milliseconds, rather than 100 microseconds - this is really important and had me confused for a while. I think it is just a font issue;

R#3-R1 (i). *We have amended the typing error.*

R#3-C2. (ii) I would be more explicit about the red / blue lasers being below / above the expected band-gap for the BTO....at the moment it is implied, but I don't think it is stated clearly in the main text at least;

R#3-R3 (ii). *We have explicitly included the energy of the blue and red photons used and the bandgap of BTO*

Action: "Now, we turn to the electrical characterization of the FTJs under illumination conducted by using blue ($\lambda = 405\text{nm}$, $E = 3.06\text{ eV}$) and red ($\lambda = 638\text{nm}$, $E = 1.94\text{ eV}$) lasers, near and well below BTO bandgap ($E_g = 3.3\text{ eV}$),⁵⁷ respectively. "

R#3-C2.(iii) while I get a sense for the vision as to how the imprint+light illuminated generation of free carriers might work, I am aware that there are other aspects to the electrostatics that are clearly there, but seem not to be so important to the light-induced switching phenomenon: perhaps a more open discussion including, for example, the role of the photo-induced carriers over and above screening available due to the conducting electrodes / free surfaces in air and to the charge-carrying point defects (oxygen vacancies) that we know are present from the ion-current characterization. In other words, can the authors explore a little more what the photo-induced carriers are able to do, in terms of screening, that other screening mechanisms that should be fully expected seem unable to facilitate.

R#0-R3 *We appreciate the advice of the referee to extend a bit the discussion of the role of photocarriers on screening. This has different answers depending on the nature of the FE material (epitaxial/polycrystalline), the nature of the interface between the FE and the electrodes (Ohmic/Schottky) and, naturally, the bandgap of the FE (photoabsorption) and the recombination/mobility of photocarriers in the FE. We have mentioned these points in the manuscript and now we have added some sentences and a couple of recent references on this issue.*

Action: We have modified the following sentence "First, the amplitude of the polarization P of a ferroelectric semiconducting single crystalline epitaxial film shall depend on the amount of available free charges²⁴ and consequently it can be modulated or eventually suppressed by internally or

externally (at the electrodes, at the ferroelectric/electrode interface or by surface adsorbates) generated photocharges.³⁷⁻⁴⁰

REVIEWER COMMENTS

Reviewer #2 (Remarks to the Author):

The authors have added more data and answered some of my question. However, not all the issues are solved and, I am afraid, that I have noticed some new points of concern.

I am glad to see the authors agree that their resistance change at V_{High} is consistent with soft breakdown. However, in the new text added in the manuscript "A coercive field, $E_c \approx 8.8$ MV/cm and an $E_{\text{imp}} \approx -2.8$ MV/cm (corresponding to $V_{\text{IMP}} \approx -1.0$ V) can be extracted. Note that E_c is below the breakdown field in polycrystalline samples.⁴⁷" "A coercive field, $E_c \approx 8.8$ MV/cm and an $E_{\text{imp}} \approx -2.8$ MV/cm (corresponding to $V_{\text{IMP}} \approx -1.0$ V) can be extracted. Note that E_c is below the breakdown field in polycrystalline samples.⁴⁷"

It is only mentioned that V_{low} is below the breakdown. That does not explicitly say that the breakdown field is similar to V_{high} and that, thus, breakdown cannot be discarded as the origin of V_{high} . Soft breakdown should be distinguished from standard ion migration that can happen at much lower voltages and does not induce "damage" in the sample. The cyclability data with 3-4 cycles, do not preclude slight damage after every cycle.

In the same line of avoiding misleading statements about the origin of the observed effects, that may set others along the wrong path, wasting resources, and may confuse the younger generations, referee 1 has brought up a very important paper I was not aware of: similar samples, but without the BTO layer (an LSMO layer on STO) have been shown to display very similar photo-induced ER [Phys. Rev. B 100, 115119 (2019)]. Therefore, if I am not missing some evidences, with the data presented so far, it seems not possible to discard that the observed effect at V_{low} does not come from the LSMO layer. In other words, according to ref. [Phys. Rev. B 100, 115119 (2019)], a 5-7% change in resistance is expected to arise from the LSMO or the LSMO/STO interface, independently of the ferroelectric character of BTO.

The reason why the authors link the change in resistance at V_{low} to ferroelectricity is the PFM phase loops showing the same switching voltages. As reviewer I have an obligation to state here that the proof for ferroelectricity in such thin layers of BaTiO₃ is not strong enough. Even when it was regularly accepted some years ago, we now know better. It is now accepted in the community that this is a very difficult problem to address and that there is no good way to distinguish ferroelectric switching from electrochemical exchange at the interfaces in such ultra-thin films. There are improved experiments one can do but an irrefutable proof is not possible with PFM alone.

One more point of concern is that the authors mention in their reply that the PFM is taken with top electrodes so that adsorbate effects are not playing a role. Unfortunately, I had not noticed this in my first reading. If so, how is it possible that they can write squares (switch the polarization locally)? Should the electric field not be uniform under the electrode?

I am not sure what other effects could explain these observation but it could be that what the authors are seeing at V_{low} is somehow an effect of ionic migration. Oxygen vacancies are associated to changes in the lattice (oxygen vacancies expand the lattice). So migration of oxygen can also lead to local deformations and, conversely, local pressure by the tip in contact-mode PFM, could induce migration of oxygen.

I am afraid that I am not able to advise publication of this manuscript.

Reviewer #3 (Remarks to the Author):

I only had minor comments to be addressed by the authors in the revised manuscript. I'm pleased to say that the authors have dealt with the points raised positively and maturely and I am happy to now recommend publication of the article.

Reviewer #2

The questions and concerns of the Reviewer #2 can be grouped as follows:

- A. Reviewer.** *The BTO films may not be ferroelectric and the observed switching of PFM contrast and ER both coinciding at VC-LOW may not necessarily be attributed to polarization switching but to other spurious effects.*

Authors.

- 1) We have succeeded on the difficult challenge of measuring the P(E) loops of the present ultrathin films. The results indisputably assess the ferroelectric character of these films. Proudly, we have included them in the main text of the revised version of the manuscript.
 - 2) The observed close coincidence of “coercive fields” measured in P(E) and PFM and experiments provide clear evidence that ER, coinciding at V_{C-LOW} , can be attributed to polarization switching.
 - 3) We emphasize that PFM images show typical fingerprints of polarization contrast rather than spurious effects.
- B. Reviewer.** *The observed photoresponsive ER could be due to the bottom electrode (LSMO)*

Authors:

- 1) We have measured the photoconductance of the LSMO films to provide new and conclusive evidence that its contribution is negligible.
- 2) The obvious changes of contrast of PFM images and ER upon illumination, that bring the polarization state of the sample to the one dictated by the imprint field, indicates that an electronic driven mechanism is at operation. The observation of these dramatic light-induced effects occur using a blue laser but not with red light strongly indicates that BTO is the relevant absorbing material. Being LSMO contribution relevant the polarization switch should occur under red and blue illumination due to LSMO's small bandgap, which is not the case.

In the following, we respond one-by-one all comments and questions.

R#2-C0 *The authors have added more data and answered some of my question. However, not all the issues are solved and, I am afraid, that I have noticed some new points of concern.*

R#2-R0 Reactions to all points are included below.

R#2-C1 *I am glad to see the authors agree that their resistance change at V_{High} is consistent with soft breakdown. However, in the new text added in the manuscript “A coercive field, $E_c \approx 8.8$ MV/cm and an $E_{imp} \approx -2.8$ MV/cm (corresponding to $V_{IMP} \approx -1.0$ V) can be extracted. Note that E_c is below the breakdown field in polycrystalline samples.⁴⁷”*

It is only mentioned that V_{low} is below the breakdown. That does not explicitly say that the breakdown field is similar to V_{high} and that, thus, breakdown cannot be discarded as the origin of V_{high} . Soft breakdown should be distinguished from standard ion migration that can happen at much lower voltages and does not induce “damage” in the sample. The cyclability data with 3-4 cycles, do not preclude slight damage after every cycle.

R#2-R1 Typically, soft-breakdown (SBD) leads to a persistent change of the conductance in metal/insulating/metal devices. Therefore, although the voltage we are applying are of the order of the SBD of polycrystalline ferroelectric⁴⁷, the observation that R(V) loops have reproducible R values upon consecutive cycles rather than decreasing, suggest that SBD may not play a prominent role.

We would like also to point out that the epitaxial nature of the film is very relevant to the dielectric response. Indeed, it is known that SBD in ferroelectric and semiconductors (including SiOx and HfOx) is mostly governed by charge flowing across the junction, rather than the voltage applied, which is mainly controlled by grain boundaries, as discussed by Stolichnov et al. (J. Applied Physics 87, 1925 (2000)). In other words, the comparison of breakdown field in polycrystalline or epitaxial films is far from obvious, and can be larger in the latter.

In any event, aiming to obtain a direct evidence of the presence or not of a SBD, we have measured the time-dependence of the current under constant electric stress as done when studying the SBD. We have selected two electrical stresses (3V and 10 V) that are close to the coercive field and the field where the observed resistance changes we attribute to (reversible) field-ionic motion, respectively. In **Fig. Res-1.** we show the time dependent measurement of the current flowing across the device under applied voltage near V_{c-low} ($= 3$ V) and near V_{c-high} ($= 10$ V). It can be observed that the current flowing under $V = 3$ V is nearly constant. Instead, the current flowing under $V = 10$ V shows a gradual reduction, with a time constant of about ≈ 1 s.. Although the time scale of the latter result might be compatible with soft breakdown, we should emphasize that we observe a reduction of current, that is an increase of the device resistance, which is opposite to the resistance decrease observed when SBD occurs in BaTiO3. ("Time-Dependent Dielectric Breakdown in BaTiO3 Thin Films, S. Desu et al. J. Electrochem. Soc., 140, L133 (1993)).

For those reasons, following the suggestion of the Ref. we accept that we cannot exclude the presence of SBD, although in our view, the available results do not strongly point in this direction.

Fig. Res-1. Time dependence of the normalized current for applied voltages near V_{c-low} ($= 3$ V) and V_{c-high} ($= 10$ V).

The Reviewer also mentioned that “cyclability data with 3-4 cycles, do not preclude slight damage after every cycle”. In **Fig. Res-2a,b** we include HRS and LRS states obtained sequentially by the application of hundreds of ± 8 V pulses ($V_{c-low} > 8$ V $< V_{c-high}$). As shown in the manuscript, at this rather voltage, ionic conduction contribution is of little relevance for ER in BTO samples and of negligible relevance on ER for the STO/BTO sample. The data collected for more than 250 cycles, demonstrates that the two resistance states remain relatively constant and are well distinguishable in both samples (Fig. Res-2b), In other words, the devices can be robustly switched. The observed variations of resistance can be ascribed to electrical noise and /mechanical instabilities.

In **Fig. Res-2c,d** we include HRS and LRS states obtained sequentially by the application of much larger voltage pulses (± 15 V) of BTO and STO/BTO samples, respectively. Notice that $V = +15$ V is well above V_{c-high} , where in our view ionic motion sets in. We observe that the junction degrades fast, after 20

cycles for BTO and after 50 for BTO/STO. This fast degradation is in agreement with the proposed ionic conduction mechanism but, a soft dielectric breakdown after several cycles, cannot be excluded.

Fig. Res-2. Endurance of HRS states and LRS states written by ± 8 V in a junction on (a) STO//LSMO/BTO/Pt sample and (b) STO//LSMO/STO/BTO/Pt sample. (c) Endurance of HRS states and LRS states written by ± 15 V in a junction on (c) STO//LSMO/BTO/Pt sample.

Reaction: We have revised the text accordingly now it reads as: “. Note that although E_C is comparable to the breakdown field in polycrystalline samples,⁴⁹ time-dependent current measurements under constant electric stress and endurance data do not agree for dielectric breakdown at $V_{C-LOW} < V_w < V_{C-HIGH}$. Instead, equivalent characterization collected for $V_w > V_{C-HIGH}$ is consistent with the proposed ionic motion scenario although the presence of soft dielectric breakdown after successive cycling cannot be disregarded (Supplementary Fig. S7 and S8).”

Regarding STO/BTO sample we have added: “and it is robust under constant cycling (Supplementary Fig. 17).”

Figures **Fig. Res-1**, **Fig. Res-2** have been included in SI, and cited in the text.

R#2-C2 In the same line of avoiding misleading statements about the origin of the observed effects, that may set others along the wrong path, wasting resources, and may confuse the younger generations, Reviewer 1 has brought up a very important paper I was not aware of: similar samples, but without the BTO layer (an LSMO layer on STO) have been shown to display very similar photo-induced ER [Phys. Rev. B 100, 115119 (2019)]. Therefore, if I am not missing some evidences, with the data presented so far, it seems not possible to discard that the observed effect at V_{low} does not come from the LSMO layer. In other words, according to ref. [Phys. Rev. B 100, 115119 (2019)], a 5-7% change in resistance is expected to arise from the LSMO or the LSMO/STO interface, independently of the ferroelectric character of BTO.

R#2-R2

The Reviewer signaled here a very interesting point. Indeed, it could happen that photon absorption in the bottom electrode (LSMO) may change its conductance and thus the resistance of the junction. Indeed, for some manganites, the occurrence of photoconductance is well documented. However, we highlight some import results against the relevant role of LSMO in our measurements:

A. Regarding the results already included in the manuscript:

- 1) The changes of the resistivity of the LSMO layer in the mentioned ref. [Phys. Rev. B 100, 115119 (2019)] are at most of 7%. As shown in Figs 3c and 4b, the changes of resistance we observe under illumination in our LSMO/BTO and LSMO/BTO/STO samples, range from 30% to 500%. Therefore, it seems that the photoconductance of the LSMO does not play a major role.

- 2) As shown in Fig 4a and 4b in the manuscript, by inserting an ultrathin (1nm) layer of STO on top of BTO, that should not modify the photoabsorption in the bottom LSMO layer, the measured ER changes by 500%, which is far from any photoconductance effect in our films (point c) above) and its orders of magnitude larger than the light promoted resistivity changes in the mentioned Reference [*Phys. Rev. B* 100, 115119 (2019)]. This observation also denies a significant role of the photoconductance of LSMO in the measured change of ER.
- B. Moreover, stimulated by the Reviewer's comment, we have performed dedicated measurements of the conductance of our LSMO film under illumination.
- 1) Direct measurements of the room-temperature resistivity of our LSMO films (see below) indicates that it is of about $\rho \approx 1 \text{ m}\Omega\cdot\text{cm}$. This value is in agreement with literature data for optimal LSMO films of similar thickness. This resistivity value implies that the series resistance contribution of LSMO to the measure device is about 75Ω . This contribution is two orders of magnitude lower than the junction resistance, indicating the small contribution of LSMO resistance to the junction resistance.

In any event we have measured the dependence of the resistivity of LSMO on illumination using the same conditions (same power and wavelength) than used in the experiments reported in the manuscript. Using a 4-probe configuration, the resistance of the bottom layer in dark and under illumination has been recorded. In **Fig. Res-3**, below we show the results. The resistivity is $\approx 1 \text{ m}\Omega\cdot\text{cm}$ in agreement in literature. Any photoresistance in our films is below noise level ($\ll 1 \text{ m}\Omega\cdot\text{cm}$). This observation excludes a contribution of LSMO photoresistance to the reported results in our manuscript.

Fig. Res-3. (a) Experimental set up to measure the LSMO film resistivity in the LSMO/BTO sample. Four needles adjacent to an array of 4 neighboring 4 Pt are used to measure in 4 probe configuration the LSMO resistivity. (b and c) I-V measurements recorded in two different configurations, obtained by permuting current and voltage probes. Data recorded in dark (black symbols) and under illumination (blue symbols) are shown in (b) and (c).

- 2) LSMO is expected to absorb red ($\lambda = 638 \text{ nm}$, $E_v = 1.94 \text{ eV}$) light due to its small bandgap of 0.6 eV. Instead, BTO is not expected to absorb light due to its greater bandgap ($E_g = 3.3 \text{ eV}$), much above $E_v = 1.94 \text{ eV}$. Thus, if LSMO has a relevant role in the polarization suppression under blue light illumination, the polarization suppression should be reproducible under red illumination. Instead, if BTO has a relevant role in the polarization suppression under blue illumination, the polarization suppression is expected to not be reproducible under red illumination. In Fig. Res-4a,b,c, PFM-phase images just after, 10 min after and after 10 min of red ($\lambda = 638 \text{ nm}$, $E_v = 1.94 \text{ eV}$, 48.5 W/cm^2) laser illumination, respectively, after electrical writing are shown. No change is observed. Therefore, the results summarized in Fig. S25a,b,c disregard important contribution from LSMO bottom electrode in the ER suppression under blue illumination.

Fig. Res-4. PFM phase contrast under red illumination. **a.** PFM phase images collected just after electrical lithography with +8 (bright) and -8V (dark). **b.** PFM phase images collected 10 min after electrical lithography with +8 (bright) and -8V (dark). for the LSMO/BTO sample. **c.** PFM phase images collected after 10 min illumination with red light red ($\lambda = 638 \text{ nm}$, $E_v = 1.94 \text{ eV}$, 48.5 W/cm^2) after electrical lithography with +8 (bright) and -8V (dark).

- 3) The change of resistance in our work is persistent as shown in Figure 6a, where no decay is observed after 5×10^4 sec delay after illumination. This observation excludes light-induced thermal effects and charge-trapping of photo induced carriers.

In summary, the ER suppression under illumination cannot be explained only by a change in the LSMO resistance under illumination.

Reaction: We have modified the main text: “These results indicate that the ferroelectric switching polarization suppression, which we recall is at the origin of the observed current peak at V_{c-low}^+ , results from photocarriers generated at the BTO layer by the blue laser. This is confirmed by the absence of polarization switching by red light, which has been observed by PFM characterization (Supplementary Fig. S13). Thermal effects, in a 4 nm BTO film, would have similar effects for both wavelengths, in contrast to observation, thus suggesting to have a minor role on the $I(V)$ photoresponse. The contribution of LSMO photoconductance has been verified, being it negligible and thus suggesting that LSMO contribution in the ferroelectric switching polarization is not relevant (Supplementary Fig. S14).”

We have included **Fig. Res-3 and 4.** in Supplementary Information.

R#2-C3 *The reason why the authors link the change in resistance at V_{low} to ferroelectricity is the PFM phase loops showing the same switching voltages. As reviewer I have an obligation to state here that the proof for ferroelectricity in such thin layers of BaTiO3 is not strong enough. Even when it was regularly accepted some years ago, we now know better. It is now accepted in the community that this is a very difficult problem to address and that there is no good way to distinguish ferroelectric switching from electrochemical exchange at the interfaces in such ultra-thin films. There are improved experiments one can do but an irrefutable proof is not possible with PFM alone.*

R#2-R3

Generally speaking, we agree with the Reviewer that the change of resistance and the PFM phase loops coinciding at the same voltages (**Fig. 3c**), may be fortuitous rather than a demonstration of ferroelectric switching in the BTO film. However, several experimental facts come to support our claim.

- 1) The PFM data were collected on nude BTO surfaces and therefore it could be reasonably speculated that the observed PFM response (phase contrast) could be originated from tip-

induced chemical reaction or charging of BTO surfaces. In contrast, the ER(V) loops were recorded on Pt electrodes, which were sputtered on the BTO surface. The experimental observation that the “free” surface of BTO in PFM measurements and the encapsulated BTO in ER(V) measurements display identical voltage-induced responses, is a strong fingerprint of its common origin, which cannot be related to adsorbates or tip induced effects, but rather to a genuine common origin. Most plausible, the polarization switching.

- 2) On the other hand, a fingerprint of charging effects or chemical reactions is the observation of a finite signal in the amplitude of the piezoelectric response at the edges of regions written with different writing voltage polarity, and a difference of the amplitude signal between these different regions and usually non-180° contrast in the phase signal piezoelectric response. Instead, a fingerprint of ferroelectric behavior is the observation of a zero signal in the amplitude of the piezoelectric response at the edges of regions written with different writing voltage polarity (i.e. different polarization sign), and no difference of the amplitude signal between these different regions and always 180° contrast in the phase signal piezoelectric response [see f.i. figure 4 in Guan et al., AIP ADVANCES 7, 095116 (2017)]. The image in **Fig. Res-5(a)** below [already included in SI as Figure S10] clearly shows the near zero piezo electric response between opposite polarity domains (see yellow arrows). The small amplitude contrast between different domains indicates the small contribution of non-ferroelectric genuine effects, which do not hide the genuine piezoelectric response. The image in **Fig. Res-5(b)** below shows 180° contrast in the phase signal piezoelectric response. Overall, although charging or chemical extrinsic effects might exist at the free surface of BTO, their presence do not deny our claim that the image strongly supports the FE character of the film.

Fig. Res-5. PFM image (a) amplitude and (b) phase of the bare surface of the BTO film, where domains of different polarity have been written by applying $V = +8V$ (bright) and $V = -8V$ (dark). See Fig S10 in supplementary information, for more details.

- 3) The I-V curves of the manuscript (Fig 3a(inset)) clearly show a distinctive jump that coincides with the phase change in PFM and disappears under blue illumination but not with the red one. This is most naturally explained as a change of P (switching), and thus ER, with blue laser illumination. Other *ad-hoc* built explanations would be probably artificial.
- 4) Additional support is provided by the direct measurement of the switching current and the corresponding P(E) loops: the classical way to assess ferroelectricity. We strength that this is an extremely challenging experiment, as the films are ultrathin (around 4 nm) and the tunnel current is large and compete with the switching current.

The extremely different resistance depending on the ferroelectric polarization direction and voltage polarity prevents to extract a reliable ferroelectric P(E) loop, from I-V curves measured with the standard configuration used in the manuscript. However, it is known top-top contacts allow mitigate this difficulty]. Consequently, we have recorded P(E) loops by antiseriial connection of two ferroelectric capacitors in the so-called top-top configuration. In this configuration, I-V asymmetry is cancelled-out, while the coercive voltage is expected to double [F. Liu et al., Sci Rep

2016, 6, 25028]. We used thin needles directly in contact with the surface to minimize the presence of leakage current added to polarization switching one. The effective contact area is $\approx 80 \mu\text{m}^2$.

Fig. Res-6. (a) I-V characteristics and (b) the corresponding P(E) loops recorded at 100 kHz on the LSMO/BTO sample. Loops are measured by connecting two top Pt adjacent electrodes L

In **Fig. Res-6** we show the P-V curves collected at 5 kHz, after leakage subtraction, for BTO and STO/BTO samples. The ferroelectric hysteresis is clear and the extracted polarization values 20 and $2 \mu\text{C}/\text{cm}^2$ are reasonable and similar to those obtained in other ultrathin BTO films [Radaelli et al., Adv. Electron. Mater. 2, 1600368 (2016)]. The reduced polarization for the STO/BTO samples is expectedly to be smaller due to different electrostatic boundary conditions of the BTO layer. For the STO/BTO sample the coercive voltage ($= 8.3 \text{ V}$) is about twice the coercive voltage of the $R(V_w)$ loop of Figure 4b ($V_{\text{imp}} = -2.2 \text{ V}$, $V_c = 4.4 \text{ V}$), as expected for the used top-top configuration. In the BTO sample, the coercive voltage ($= 4.8 \text{ V}$) is larger as expected, than the coercive voltage observed in the $R(V_w)$ loop of Figure 2b ($V_{\text{imp}} = -1.0 \text{ V}$, $V_c = 3.5 \text{ V}$), although it is somewhat smaller than expectations ($\approx 7 \text{ V}$). The imprint voltage in the P-V loops is residual as expected due to the symmetric top-top configuration used ($V_{\text{imp}} = 0.1$ and 0.2 V for BTO and STO/BTO films, respectively).

Reaction:

- We have included “The observation of near-zero piezo electric response and 180° phase shift between stable domains of opposite polarity in PFM images (Supplementary Fig. S3 and S4) strongly supports their ferroelectric nature.^{47, 48}”
- We have included Fig. Res-6. (a,b) in the Fig. 3 and 4 of the manuscript, respectively.
- We have added a Supplementary Fig. S25 describing the leakage subtraction procedure.
- Methods section has been expanded with the description of the P-V top-top measurements.
- We have included in the text for the BTO sample: “P(V) and I(V) loops (Fig. 3d) collected in top-top configuration (see Methods) further indicate the ferroelectric nature of the sample. Switching current peaks appear at 5.6 V, which is about twice V_c observed in the $R(V_w)$ loop, as expected from an antiseriial connection of ferroelectric capacitors.⁵⁰ Polarization value is also in reasonable agreement with bulk values and similar to values obtained in BTO films grown in the same nominal conditions.⁵¹”
- We have included in the text for the STO/BTO sample: “P(V) and I(V) loops (Fig. 4d) collected in top-top configuration (see Methods) further indicate the ferroelectric nature of the sample. Switching current peaks appear at 4.4 V, which is close to two times V_c ($\approx 8.4 \text{ V}$) from $R(V_w)$ loop, as expected.⁵⁰ The reduced polarization compared with the LSMO/BTO sample is expectedly smaller due to different electrostatic boundary conditions of the BTO layer.”

R#2-C4 *One more point of concern is that the authors mention in their reply that the PFM is taken with top electrodes so that adsorbate effects are not playing a role. Unfortunately, I had not noticed this in my first reading. If so, how is it possible that they can write squares (switch the polarization locally)? Should the electric field not be uniform under the electrode?*

R#2-R4. We guess that there is a confusion here. In our reply to Reviewers letter, in the section R#2-R2 we wrote. *“We strength that ER data were collected using electrodes and thus surface adsorbates, whose presence is unavoidable in PFM, do not play a major role. The observed coincidence of switching voltages could hardly be a coincidence.”*

The PFM measurements were done without electrodes, whereas electrodes where used in the ER measurements.

R#2-C5 *I am not sure what other effects could explain these observation but it could be that what the authors are seeing at V_{low} is somehow an effect of ionic migration. Oxygen vacancies are associated to changes in the lattice (oxygen vacancies expand the lattice). So migration of oxygen can also lead to local deformations and, conversely, local pressure by the tip in contact-mode PFM, could induce migration of oxygen.*

R#2-R5. The suggestions of the Reviewer are interesting but at present, to our view, would be speculative. To the best of our knowledge, there are no available data in the literature, to support ionic migration induced by the pressure of the tip, leading to a unique non-elastic ionic motion mimicking a voltage induced piezo response

In any event. This speculation however would require also to address the obvious effect of blue laser on the PFM and ER response. This is far from obvious.

R#2-C6 *I am afraid that I am not able to advise publication of this manuscript.*

R#2-R6 In our opinion, the reasonable objections indicated by the Reviewer are fundamentally refuted by the experimental data.

Reviewer #3 (Remarks to the Author):

R#3-C1 *I only had minor comments to be addressed by the authors in the revised manuscript. I'm pleased to say that the authors have dealt with the points raised positively and maturely and I am happy to now recommend publication of the article.*

R#3-R1 We are thankful to the Reviewer for his/her positive appreciation on the detailed answer we provided to all questions and comments and for the recommendation to publish.

REVIEWERS' COMMENTS

Reviewer #2 (Remarks to the Author):

The authors have answered all and each of my concerns. They have performed a series of experiments that were needed and I am grateful and convinced of the validity of their claims. I can now advise publication of this manuscript in Nature Communications.